# ITFormer: Bridging Time Series and Natural Language for Multi-Modal QA with Large-Scale Multitask Dataset

Yilin Wang [1 2]   Peixuan Lei [1 *]   Jie Song [1 *]   Yuzhe Hao [1 *]   Tao Chen [1 *]   Yuxuan Zhang [1]
Lei Jia [1]   Yuanxiang Li [1]   Zhongyu Wei [2 3]

## Abstract

Time-series data are critical in diverse applications, such as industrial monitoring, medical diagnostics, and climate research. However, effectively integrating these high-dimensional temporal signals with natural language for dynamic, interactive tasks remains a significant challenge. To address this, we introduce the Time-Series Question Answering (Time-Series QA) task and release EngineMT-QA, the first large-scale, multitask, temporal-textual QA dataset designed to capture complex interactions between time-series signals and natural language. Building on this resource, we propose the Instruct Time Transformer (ITFormer), a novel framework that bridges time-series encoders with frozen large language models (LLMs). ITFormer effectively extracts, aligns, and fuses temporal and textual features, achieving a strong improvement in QA accuracy over strong baselines with fewer than 1% additional trainable parameters. By combining computational efficiency with robust cross-modal modeling, our work establishes a adaptable paradigm for integrating temporal data with natural language, paving the way for new research and applications in multi-modal AI. More details about the project, including datasets and code, are available at: https://pandalin98.github.io/itformer_site/.

## 1. Introduction

Time-series data have become increasingly significant in numerous real-world applications(Wang et al., 2025a), such

as industrial condition monitoring(Wang et al., 2025b) (e.g., turbine engine health management)(Wang et al., 2022) , medical diagnostics (e.g., electrocardiogram analysis) (Yi et al., 2024; Gui et al., 2024), and climate research (Zhang et al., 2023; Wu et al., 2023; Bi et al., 2023). These data often exhibit high-dimensional patterns and intricate temporal dependencies(Wang et al., 2023; 2024c), carrying valuable information essential for precise analysis and decision-making. While existing research has made progress in tasks like time-series classification(Yue et al., 2022; Goswami et al., 2024), forecasting(Zhou et al., 2022; Zhang & Yan, 2023; Liu et al., 2024d), and anomaly detection (Jin et al., 2024a; Darban et al., 2025), most efforts remain focused on single-modality tasks, limiting their applicability to dynamic and interactive environments.

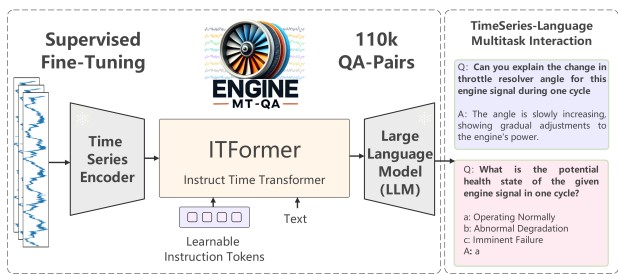

**Figure 1:** EngineMT-QA: A large-scale QA dataset based on aero engine time-series signals. The proposed ITFormer framework seamlessly connects any time-series encoder with LLMs, modeling temporal-text interaction by embedding time-series semantics into natural language.

A growing need exists for practical applications where users can interact with time-series data directly using natural language. For example, a engineer might ask if an engine component shows abnormal vibration during a specific time period or request a summary of performance trends to inform maintenance decisions. This demand lies at the intersection of time-series analytics and natural language processing (NLP), an area gaining relevance with the advancements in large language models (LLMs) (Wei et al., 2022). However, despite increasing interest in cross-modal research (Sun et al., 2024; Rasheed et al., 2024; Lu et al., 2024), there has been limited effort to systematically explore the

*Equal contribution [1]School of Aeronautics and Astronautics, Shanghai Jiao Tong University, Shanghai, China [2]Shanghai Innovation Institute, Shanghai, China [3]School of Data Science, Fudan University, China. Correspondence to: Yuanxiang Li < yuanxli@sjtu.edu.cn>.

*Proceedings of the 42nd International Conference on Machine Learning*, Vancouver, Canada. PMLR 267, 2025. Copyright 2025 by the author(s).

complex and abstract semantics underlying time-series data in the context of natural language queries. Moreover, no standardized benchmark currently exists to evaluate methodologies in Time-Series Question Answering (Time-Series QA), leaving this domain underexplored.

To address this gap, we introduce the Aero Engine Multi-Task Question Answering Dataset (EngineMT-QA), a large-scale, multi-task, and temporal-textual QA dataset specifically designed for time-series data. EngineMT-QA is constructed based on real-world aero engine operational and maintenance scenarios, aiming to capture the intricate interactions between time-series signals and natural language. The dataset consists of over 110k question-answer pairs derived from 33 sensor channels, covering a diverse set of queries across four key tasks: understanding, perception, reasoning, and decision-making. These tasks are designed to reflect the inherent complexity of time-series data and explore their underlying semantic structures. For instance, some questions assess the ability to interpret subtle amplitude variations in specific signals and their semantic implications, while others require reasoning about the probability of component failure and making informed decisions on whether to proceed with repairs or continue operation based on the deeper semantic meaning of the time-series signal. By releasing this dataset, we aim to facilitate research on bridging time-series data with natural language, paving the way for more intelligent and interpretable solutions in real-world applications.

Building on this new data resource, we introduce Instruct Time Transformer (ITFormer), a novel framework designed to bridge temporal sequence data and frozen LLMs for efficient temporal-textual multimodal interactions in Fig. 1. ITFormer acts as an intermediary connector, enabling seamless integration between temporal encoders and frozen LLMs by addressing three core challenges. First, ITFormer employs Time Token Position Encoding to systematically represent temporal, channel, and segment-level position information. Second, ITFormer introduces Learnable Instruct Tokens and Instruct Time Attention to dynamically align and fuse temporal features with task-specific textual queries, creating a unified semantic space for effective modeling. Third, the Time Token as Language strategy represents temporal features as language-compatible tokens, enabling smooth integration with LLMs while requiring minimal computational overhead.

Empirical evaluations demonstrate that ITFormer achieves state-of-the-art performance, significantly surpassing strong baselines in QA tasks. By leveraging fewer than 1% additional trainable parameters, ITFormer balances computational efficiency with robust performance, advancing temporal-textual cross-modal modeling.

In summary, our contributions are follows:

1. We formally introduce the Time-Series QA task and release the first large-scale, multi-task, temporal-textual QA dataset, EngineMT-QA, which enables systematic exploration of temporal-textual interactions.

2. We propose ITFormer, an innovate framework for aligning and fusing time-series representations with natural language, achieving substantial performance improvements across diverse QA tasks with minimal training parameters. ITFormer's core methods include Time Token Position Encoding, Learnable Instruct Tokens, Instruct Time Attention, and Time Token as Language, which collectively drive effective cross-modal modeling.

3. We establish a robust and adaptable paradigm for integrating complex time-series tasks into end-to-end QA frameworks, paving the way for more comprehensive multi-modal applications and offering valuable benchmarks and tools to the research community.

## 2. Related Work

**Time-Series Analysis**    Time-series research has traditionally focused on tasks such as classification, forecasting, and anomaly detection (Hamilton, 2020). Early methods like ARIMA (Shumway et al., 2017) and VAR (Clements & Mizon, 1991) were designed to model linear temporal dependencies. With the advent of deep learning, neural models(Dennis et al., 2019; Oreshkin et al., 2022) significantly improved the ability to capture non-linear patterns in time-series data. More recently, Transformer-based architectures have pushed the boundaries of long-range dependency modeling(Zhou et al., 2021; 2022; Nie et al., 2023; Liu et al., 2024b), leading to the development of foundational time-series models that solve various tasks (Liu et al., 2024d; Gao et al., 2024). However, these foundational models are predominantly single-modal, limiting their applicability in scenarios that require multi-modal integration or interactive analysis.

**Large Language Models and Multi-Modal QA**    Large Language Models (LLMs) have demonstrated remarkable proficiency in various NLP tasks, including machine translation (Vaswani, 2017), text summarization (See et al., 2017), and question answering (Rajpurkar, 2016). By training on massive text corpora, they capture rich linguistic patterns that generalize across diverse downstream applications. Recent advancements have extended LLMs to multimodal tasks, primarily in the vision-language domain (Lu et al., 2019; Li et al., 2022), where models integrate textual and visual information to improve reasoning and comprehension. Similarly, the field of multi-modal QA has evolved around vision-language benchmarks such as VQA (Antol et al., 2015) and Visual Dialog (Das et al., 2017), which

require models to fuse image features with textual queries to generate accurate answers (Tan & Bansal, 2019; Lu et al., 2019). While these efforts have driven research in cross-modal alignment techniques, the integration of time-series data into LLMs remains largely unexplored. Unlike visual inputs, time-series data pose unique challenges, as their semantics are more abstract and governed by multi-scale temporal dependencies and continuous signal dynamics. Moreover, the vast availability of time-series data and the increasing need for interactive analysis necessitate more fine-grained research on temporal-textual multimodal modeling beyond conventional multimodal QA frameworks.

**Towards Time-Series QA** Although some efforts have attempted to integrate text and time-series modalities(Jin et al., 2024c;b), existing works predominantly treat textual inputs as auxiliary features to enhance traditional time-series tasks, such as prediction(Wang et al., 2024b) or classification (Wang et al., 2024a). These approaches often focus on pre-defined prompt or rely on simplified datasets, limiting their ability to generalize to real-world scenarios. While some efforts have explored natural language interfaces for time-series data (Liu et al., 2024a), they primarily focus on generating narrative descriptions or extracting basic patterns (Wang et al., 2024a), without delving into deeper semantic modeling or capturing the complex interactions between time-series data and language. To the best of our knowledge, there is no systematic exploration of *Time-Series QA*, where models are required to directly infer answers from raw temporal signals guided by rich natural language queries. The absence of a standardized benchmark further limits progress in this domain. To bridge this gap, we introduce **EngineMT-QA**, a dedicated dataset for time-series question answering, alongside our novel architecture, **ITFormer**.

## 3. Problem Definition

Time-series analysis encompasses a diverse range of tasks, including understanding, perception, reasoning, and decision-making, among other high-level cognitive processes. These tasks involve diverse input-output formats, making it challenging to establish a unified framework. To address this, we propose a question-answering (QA) paradigm that reformulates all time-series tasks into a single, coherent framework. Formally, given a time-series collection $\mathcal{T} = \{\mathcal{T}_1, \mathcal{T}_2, \ldots, \mathcal{T}_N\}$, where each segment $\mathcal{T}_i = \{x_{i,1}, x_{i,2}, \ldots, x_{i,L_i}\}$ consists of sensor signals across $V$ channels with $L_i$ steps, i.e., $x_{i,t} \in \mathbb{R}^V$, and a natural language query $q \in \mathcal{Q}$, the objective is to generate a corresponding natural language response $a \in \mathcal{A}$. This task can be formulated as learning a unified mapping function:

$$f : (\mathcal{T}, q) \mapsto a, \qquad (1)$$

where the mapping function $f$ encapsulates the interaction between the time series and natural language modalities. $f$ is decomposed into several components to facilitate effective modeling. First, the time series $\mathcal{T}$ is encoded into a latent representation $\mathcal{H}_T$ via a time series encoder $\Phi_T$, i.e., $\mathcal{H}_T = \Phi_T(\mathcal{T})$. Simultaneously, the question $q$ is transformed into a semantic representation $\mathcal{H}_q$ via a language encoder $\Phi_q$, i.e., $\mathcal{H}_q = \Phi_q(q)$. These modality-specific representations are then fused through an interaction mechanism $\Psi$ to obtain a joint representation $\mathcal{H}_{\text{fusion}}$, i.e., $\mathcal{H}_{\text{fusion}} = \Psi(\mathcal{H}_T, \mathcal{H}_q)$. Finally, the fused representation $\mathcal{H}_{\text{fusion}}$ is decoded into the answer $a$ under the condition of the question $q$. The decoding process is defined as:

$$a = \Phi_a(\mathcal{H}_{\text{fusion}} \mid q), \qquad (2)$$

where $\Phi_a$ represents the decoder function that incorporates the alignment and fusion results $\mathcal{H}_{\text{fusion}}$ alongside the textual question $q$ to generate the final answer.

## 4. Methodology

We propose **ITFormer** (Fig. 2), a novel framework that enables efficient temporal-textual multimodal interactions by bridging time-series data and frozen LLMs. Acting as an intermediary, ITFormer seamlessly integrates any time-series encoder with any frozen LLM, extracting the semantic essence of temporal tokens based on task-specific instructions for alignment and fusion in a shared semantic space.

### 4.1. Instruct Time Transformer

Designed to address challenges in temporal-textual alignment, such as modality heterogeneity and computational efficiency, ITFormer incorporates key components to achieve effective representation fusion.

**Time Token Position Encoding (TPE)** To enable the model to effectively distinguish between multi-segment and multi-dimensional temporal tokens, we design a position encoding mechanism that operates at three levels: temporal steps, channel semantics, and temporal segments. Specifically, the temporal tokens are first processed by the time series encoder $\Phi_T$, and position encodings are subsequently added to the resulting features. The final representation is given by:

$$\mathcal{H}_T = \Phi_T(\mathcal{T}) + \mathbf{P}_{\text{time}} + \mathbf{P}_{\text{channel}} + \mathbf{P}_{\text{segment}}, \qquad (3)$$

The term $\Phi_T(\mathcal{T})$ refers to the initial feature representation of the time series $\mathcal{T}$, produced by the time series encoder. For a given segment $\mathcal{T}_i = \{x_{i,1}, x_{i,2}, \ldots, x_{i,L_i}\}$, where

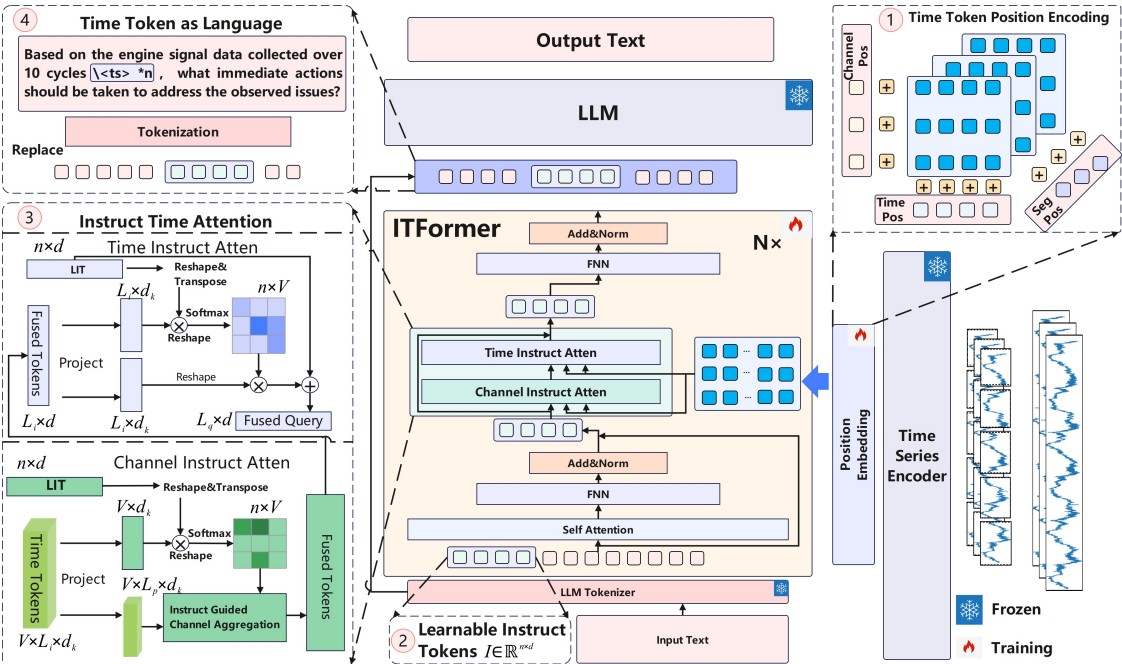

**Figure 2:** The design of ITFormer includes several key components: *1. Time Token Position Encoding (TPE)*, *2. Learnable Instruct Tokens (LIT)*, and the *3. Instruct Time Attention (ITA)* mechanism. The framework also incorporates the *4. Time Token as Language (TAL)* strategy, which represents temporal semantics as language tokens to enhance the expressiveness of fused representations.

$x_{i,t} \in \mathbb{R}^V$, the encoder produces a latent representation $\Phi_T(\mathcal{T}_i) \in \mathbb{R}^{L'_i \times V \times d}$, where $L'_i$ represents the number of tokens in the temporal dimension after the Patch transformation (Nie et al., 2023). To incorporate positional information, $\mathbf{P}_{\text{time}}$ employs sinusoidal encodings along the temporal steps $t = 1, 2, \ldots, L'_i$, capturing the sequential structure of the time series. In parallel, $\mathbf{P}_{\text{channel}}$ introduces learnable embeddings to differentiate semantic information across the $V$ channels. For multiple segment inputs, token sequences are concatenated as $\Phi_T(\mathcal{T}) \in \mathbb{R}^{L' \times V \times d}$, where $L' = \sum_{i=1}^{N} L'_i$. The encoding $\mathbf{P}_{\text{segment}}$ applies rotary encodings to distinguish between temporal segments $\{\mathcal{T}_1, \mathcal{T}_2, \ldots, \mathcal{T}_N\}$, ensuring clear identification in multi-segment scenarios.

Together, these position encodings enable the model to leverage the hierarchical structure of temporal data, thereby facilitating effective alignment and seamless interaction with textual semantics.

**Learnable Instruct Tokens (LIT)** To facilitate the alignment between temporal and textual modalities, we introduce LIT tokens, denoted as $\mathbf{I} = \{\mathbf{i}_1, \mathbf{i}_2, \ldots, \mathbf{i}_n\}$, which are used to extract and integrate the semantics from both language and time-series signals. These tokens are prepended to the semantic vector representation of the natural language query $q \in \mathbb{R}^{L_q \times d}$. Here, $q$ is obtained by passing the raw textual query through the tokenizer of the frozen LLM. The

augmented query is thus represented as:

$$\tilde{q} = [\mathbf{I}; q], \qquad (4)$$

where $\mathbf{I} \in \mathbb{R}^{n \times d}$ shares the same embedding dimensionality as $q$ to ensure compatibility. The concatenated sequence $\tilde{q}$ undergoes a self-attention mechanism to refine its semantic representation, resulting in:

$$\mathcal{H}_q = \text{SelfAttention}(\tilde{q}), \qquad (5)$$

where $\mathcal{H}_q \in \mathbb{R}^{(n+L_q) \times d}$ fused the task-specific semantics of $q$ in $\mathbf{I}$. After the self-attention operation, the first $n$ tokens of $\mathcal{H}_q$ are extracted as the final output, denoted as $\mathbf{I}^* = \{\mathbf{i}_1^*, \mathbf{i}_2^*, \ldots, \mathbf{i}_n^*\}$, where:

$$\mathbf{I}^* = \mathcal{H}_q[:n]. \qquad (6)$$

LIT serve as condensed semantic instructions, encoding task-relevant information from query to guide downstream alignment and interaction between temporal and textual representations. This design ensures that the alignment process leverages both task-specific guidance and the contextual semantics of the textual query, enhancing the fusion quality in the shared semantic space.

**Instruct Time Attention (ITA)** The ITA mechanism employs a two-stage process to dynamically align and fuse temporal and textual representations. This involves *Channel*

*Instruct Fusing* followed by *Time Instruct Attention*, leveraging the task-specific LIT to guide the alignment. ITA first aggregates relevant features of temporal tokens along the feature dimension based on the instruction, capturing the intrinsic semantics of each temporal step. Subsequently, the mechanism aggregates these features across the temporal dimension to capture the semantic evolution over time.

*Channel Instruct Fusing*

In this stage, the temporal features $H_T \in \mathbb{R}^{L' \times V \times d}$, where $L'$ represents the number of temporal tokens, $V$ the number of channels, and $d$ the feature dimension, are aggregated along the channel dimension under the guidance of the LIT $\mathbf{I}^*$. This step extracts channel-wise information most relevant to the LIT at each temporal step.

The attention mechanism is defined as:

$$\mathbf{Q}_{\text{channel}} = \mathbf{I}^* \mathbf{W}_q, \mathbf{K}_{\text{channel}} = \mathcal{H}_T \mathbf{W}_k, \mathbf{V}_{\text{channel}} = \mathcal{H}_T \mathbf{W}_v, \quad (7)$$

where are learnable projection matrices.

The channel-wise attention weights are computed as:

$$\mathbf{A}_{\text{channel}} = \text{Softmax}\left(\frac{\mathbf{Q}_{\text{channel}}\mathbf{K}_{\text{channel}}^{\top}}{\sqrt{d_k}}\right), \quad (8)$$

where represents the relevance of each channel to the LIT.

Using these attention weights, the channel-wise aggregation is performed as:

$$\mathcal{H}_{\text{channel}}[l, k] = \frac{1}{n}\sum_{q=1}^{n}\sum_{v=1}^{V}\mathbf{A}_{\text{channel}}[q, v] \cdot \mathbf{V}_{\text{channel}}[l, v, k], \quad (9)$$

The resulting represents the fused tokens $\mathcal{H}_{\text{channel}} \in R^{L' \times d}$ for each temporal step.

*Time Instruct Attention* The output from the previous stage, $\mathcal{H}_{\text{channel}}$, is further refined to capture temporal interactions guided by the LIT. This stage computes cross-modal attention between the temporal sequence $\mathcal{H}_{\text{channel}}$ and the LIT $\mathbf{I}^*$. The attention mechanism is defined as:

$$\mathbf{Q}_{\text{time}} = \mathbf{I}^* \mathbf{W}_q', \mathbf{K}_{\text{time}} = \mathcal{H}_{\text{channel}} \mathbf{W}_k', \mathbf{V}_{\text{time}} = \mathcal{H}_{\text{channel}} \mathbf{W}_v', \quad (10)$$

where $\mathbf{W}_q', \mathbf{W}_k', \mathbf{W}_v' \in \mathbb{R}^{d \times d_k}$ are learnable projection matrices. The time-wise attention is computed as:

$$\mathcal{H}_{\text{fusion}} = \text{Softmax}\left(\frac{\mathbf{Q}_{\text{time}}\mathbf{K}_{\text{time}}^{\top}}{\sqrt{d_k}}\right)\mathbf{V}_{\text{time}}, \quad (11)$$

where $\mathcal{H}_{\text{fusion}} \in \mathbb{R}^{n \times d}$ represents the fused temporal representation aligned with the textual LIT.

**Time Token as Language (TAL)** The fused temporal representation $\mathcal{H}_{\text{fusion}} \in \mathbb{R}^{n \times d}$ is treated as a sequence of task-specific tokens and directly aligned with the natural language query $\mathcal{H}_q \in \mathbb{R}^{L_q \times d}$ by replacing specific placeholder

tokens in the query. This operation ensures that the temporal semantics encoded in $\mathcal{H}_{\text{fusion}}$ are seamlessly integrated into the query's semantic structure. The augmented input sequence is defined as:

$$\tilde{\mathcal{H}}_q = \text{ReplacePlaceholders}(\mathcal{H}_q, \mathcal{H}_{\text{fusion}}), \quad (12)$$

where ReplacePlaceholders$(\cdot, \cdot)$ denotes the replacement of predefined placeholder tokens in $\mathcal{H}_q$ with the corresponding tokens from $\mathcal{H}_{\text{fusion}}$. The resulting sequence $\tilde{\mathcal{H}}_q \in \mathbb{R}^{(L_q + n) \times d}$ aligns temporal and textual semantics within the same embedding space, making it suitable for processing by the frozen LLM. The decoder in the frozen LLM then generates the answer $a$ as:

$$a = \Phi_a(\tilde{\mathcal{H}}_q), \quad (13)$$

where $\Phi_a$ represents the decoding process in the LLM, utilizing the integrated temporal-textual representation $\tilde{\mathcal{H}}_q$ to produce the final answer.

## 4.2. Training Process

ITFormer is trained via supervised fine-tuning (SFT) by minimizing the cross-entropy loss between generated and ground-truth answers. Only ITFormer's parameters are updated, while the time encoder $\Phi_T$ and LLM $\Phi_q$ (e.g., 7B parameters) remain frozen. The time encoder $\Phi_T$ extracts temporal embeddings $\mathcal{H}_T$ from multivariate time-series data $\mathcal{T} \in \mathbb{R}^{L \times V}$:

$$\mathcal{H}_T = \Phi_T(\mathcal{T}). \quad (14)$$

**Alignment Training** Only the alignment module $\Psi$ (approximately 0.07% of total parameters) is updated, projecting $\mathcal{H}_T$ into the semantic space of query embeddings $\mathcal{H}_q = \Phi_q(q)$ to form the fused representation:

$$\mathcal{H}_{\text{fusion}} = \Psi(\mathcal{H}_T, \mathcal{H}_q). \quad (15)$$

The frozen LLM decoder $\Phi_q$ then generates the final answer $a$:

$$a = \Phi_q(\mathcal{H}_{\text{fusion}}). \quad (16)$$

The training minimizes the cross-entropy loss:

$$\mathcal{L}_{\text{align}}(\theta_\Psi) = -\sum_{(\mathcal{T}, q, a) \in \mathcal{D}} \log P(a \mid \mathcal{T}, q; \theta_\Psi), \quad (17)$$

where only $\theta_\Psi$ is updated, ensuring efficient multimodal alignment by leveraging pretrained components for robust cross-modal QA performance.

## 5. Experiment

### 5.1. Dataset Description

We constructed a large-scale, complex, multi-task QA dataset, EngineMT-QA, based on the N-CMAPSS(Arias Chao et al., 2021) aero engine dataset,

reflecting real-world engine operation and maintenance scenarios. The dataset construction process is detailed in Appendix A. EngineMT-QA comprises 11k QA pairs, designed to cover four key task categories: Understanding, Perception, Reasoning, and Decision-Making. Detailed statistical information can be found in Appendix B.

The Understanding task requires the model to interpret the relationships among various sensors and derive the semantic implications of changes in sensor signals. The Perception task focuses on uncovering the underlying health state semantics of the signals, diagnosing faults, and identifying root causes. The Reasoning task challenges the model to infer degradation trends by analyzing semantic variations across 10 consecutive cycles and to predict both the probability of future failures and the remaining useful life of the engine. Finally, the Decision-Making task involves making end-to-end operational and maintenance decisions, such as determining whether and when to repair potential components, based on the reasoning capabilities of the model.

Evaluation methodologies are task-specific. For the Understanding and Decision-Making tasks, the answers are open-ended and evaluated using BLEU(Papineni et al., 2002) and Rouge-L(Lin, 2004) to assess the quality of generated responses. In contrast, the Perception and Reasoning tasks have fixed answers and are evaluated using Accuracy and F1 scores to measure model performance. As shown in Fig. 3, EngineMT-QA unifies diverse tasks within complex temporal scenarios into a cohesive TimeSeries-QA framework, more QA examples can be found in Appendix C. This dataset provides a robust foundation for constructing and evaluating temporal-textual multimodal models capable of effectively handling intricate real-world scenarios.

### 5.2. Main Results

In the main experiments, we use a 4-layer, 8-head PatchTST (Nie et al., 2023) as the time-series encoder, with 10-minute (600 points) time-series segments of 33 channel engine signals. The patch length and stride are set to 60, and pretraining runs for 10 epochs. ITFormer is a 2-layer, 8-head model, integrated with the frozen qwen2.5 (Yang et al., 2024) series LLMs. The supervised fine-tuning (SFT) stage lasts 2 epochs, and all training is conducted on 4 NVIDIA H100 GPUs.

For evaluation, we compare against multimodal APIs (ChatGPT-4o (OpenAI, 2024) and Gemini (Team et al., 2023), with prompt templates in Appendix D), time-series-text models (Time-LLM (Jin et al., 2024b) and AutoTime (Liu et al., 2024c), adapted to text space), and vision-text models (InstructBlip (Dai et al., 2023), MCAN-VQA (Zhou Yu et al., 2019), and CoCa (Yu et al., 2022), where the image encoder is replaced with the same PatchTST encoder). In all comparison experiments, ITFormer models

were trained on the EngineMT-QA dataset, with training conducted on the training subset and evaluation on the test subset. For a fair comparison, existing multimodal models like GPT-4o and Gemini, vision-text models like Instruct-Blip, MCAN-VQA, and CoCa, and time-series-text models like Time-LLM and AutoTime were all adapted to use time-series encoders instead of their original image encoders. Specifically, the PatchTST time-series encoder, which is identical to the one used in ITFormer, replaced the image encoder in these vision-text models. This ensures that all models are evaluated under the same conditions, using time-series data in place of images. The training steps and epochs were consistent across all models, ensuring that the performance differences between ITFormer and the comparison models are due to the architecture and not variations in training configurations.

The experimental results in Table 1 demonstrate that **ITFormer achieves state-of-the-art performance across all four tasks** in EngineMT-QA. The largest model, **ITFormer-7B**, consistently achieves the best performance on every metric, including Rouge-L and BLEU for open-ended tasks (Understanding: **58.04/38.23**; Decision: **56.62/38.68**), as well as Accuracy and F1 for classification tasks (Perception: **65.07/68.36**; Reasoning: **88.69/88.69**). These results demonstrate that ITFormer effectively handles both generative and discriminative temporal-textual QA.

Performance scales well with model size: **ITFormer-3B** ranks second across all tasks, while **ITFormer-0.5B** already surpasses strong baselines such as InstructBlip, CoCa, Time-LLM, and AutoTime. This confirms the scalability and efficiency of our design, which integrates time-series encoders with frozen LLMs through lightweight temporal-textual alignment modules.

Among baselines, vision-language models like InstructBlip and MCAN-VQA perform reasonably well on generative tasks but exhibit less stability across classification tasks. Time-series-text models like Time-LLM and AutoTime underperform in Decision-making due to limited generative capacity. Multimodal APIs such as ChatGPT-4o and Gemini achieve low scores across all tasks (e.g., ChatGPT-4o Decision BLEU: 3.30; Reasoning F1: 39.53), highlighting their lack of specialization for structured multivariate temporal signals.

The performance gains of ITFormer can be attributed to its dedicated components—**TPE, LIT, ITA, and TAL**—which enable hierarchical temporal encoding, modality alignment, and efficient signal-to-language fusion. Ablation results show that **TPE** and **ITA** contribute the most individually, while combining all four components leads to the highest overall performance, validating their complementary nature.

Overall, ITFormer offers a unified framework for complex

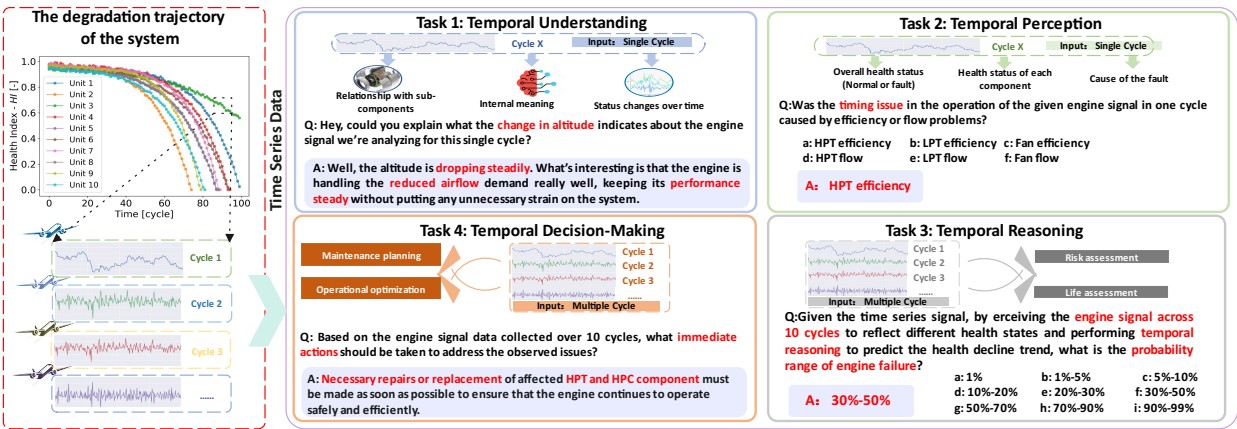

**Figure 3:** Overview of the EngineMT-QA dataset. This dataset comprises 11k QA pairs across four tasks: Understanding, Perception, Reasoning, and Decision-Making. Each task reflects specific operational requirements of engine maintenance, including sensor relationships, health state interpretation, trend prediction, and maintenance decision-making.

time-series QA, delivering robust performance with minimal parameter tuning, and setting a strong baseline for temporal-textual reasoning in real-world scenarios.

### 5.3. Adaptive Study

To evaluate the capability of ITFormer in effectively bridging diverse temporal encoders and LLMs, we conducted a series of experiments aiming to transform various time-series encoders and LLMs into a unified and robust multimodal framework. Specifically, we examined ITFormer's adaptability by integrating multiple LLMs (e.g., Qwen, LLaMA (Touvron et al., 2023), and GLM (Du et al., 2022)) with different temporal encoders (e.g., PatchTST, Informer, and Crossformer). The pretraining and subsequent SFT settings of the Time-series Encoder remain consistent with those in the main experiment. The results (Fig. 4) demonstrated that ITFormer consistently maintained stable performance across all combinations, achieving competitive results in terms of both average Accuracy and Rouge-L scores. Furthermore, a notable trend was observed: as the scale of the LLM increased, the performance of the combined model improved correspondingly. This indicates that larger LLMs may provide ITFormer with a more favorable optimization landscape during the supervised fine-tuning stage, enhancing its overall performance.

### 5.4. Ablation Study

We conducted ablation studies to investigate the impact of ITFormer layer numbers and LIT lengths on model performance. As shown in Fig. 5, a 2-layer ITFormer achieves optimal performance, balancing computational efficiency and accuracy, while additional layers yield diminishing returns under current data conditions, likely due to optimization challenges. Similarly, experiments on LIT lengths demon-

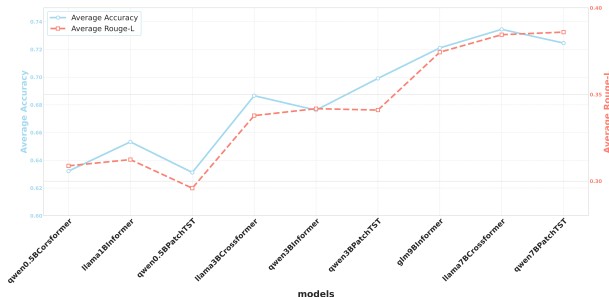

**Figure 4:** ITFormer maintains robust performance across architectures, with larger LLMs enhancing optimization and improving results.

strate that excessively short tokens fail to capture sufficient multimodal information, whereas overly long tokens result in sparse and less effective feature extraction. The results indicate that an LIT length of 25 tokens strikes an ideal balance, enabling efficient and robust temporal-textual multimodal integration.

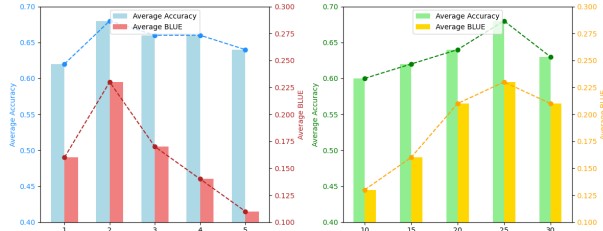

**Figure 5:** Impact of ITFormer layers and LIT length on model performance. A moderate number of ITFormer layers optimizes performance, while very short LIT lengths limit multimodal extraction, and overly long lengths lead to sparse information.

To evaluate the contribution of each component in ITFormer, we conduct an ablation study on four key modules: the Instruct Time Attention (ITA), Time Token Position Encoding

**Table 1:** Performance table for the four tasks in EngineMT-QA, with the best and second-best performances highlighted in orange and red, respectively.

| Method | Method Type | Understanding | | Perception | | Reasoning | | Decision | |
|---|---|---|---|---|---|---|---|---|---|
| | | Rouge-L↑ | BLUE↑ | Accuracy↑ | F1↑ | Accuracy↑ | F1↑ | Rouge-L↑ | BLUE↑ |
| ChatGPT-4o | Multimodal API | 15.23 | 7.69 | 52.14 | 46.91 | 29.32 | 39.53 | 9.19 | 3.30 |
| Gemini | Multimodal API | 21.81 | 6.43 | 52.11 | 40.01 | 22.04 | 38.42 | 8.80 | 2.90 |
| InstructBlip | Vision-Text | 47.31 | 25.29 | 53.11 | 51.61 | 51.64 | 51.64 | 31.21 | 14.47 |
| MCAN-VQA | Vision-Text | 48.52 | 26.59 | 59.79 | 54.91 | 54.78 | 54.78 | 31.90 | 14.92 |
| CoCa | Vision-Text | 46.21 | 25.12 | 52.21 | 50.11 | 50.21 | 50.21 | 33.92 | 13.68 |
| Time-LLM | Time-Series-Text | 29.40 | 18.60 | 46.05 | 46.97 | 47.54 | 47.54 | 28.10 | 11.51 |
| AutoTime | Time-Series-Text | 27.24 | 17.99 | 43.45 | 46.47 | 49.96 | 49.97 | 23.55 | 9.80 |
| ITFormer-0.5B | Proposed Method | 49.36 | 28.05 | 63.95 | 54.39 | 80.18 | 80.09 | 31.23 | 15.26 |
| ITFormer-3B | Proposed Method | **54.37** | **33.44** | **64.07** | **58.36** | **83.30** | **83.22** | **44.67** | **24.37** |
| ITFormer-7B | Proposed Method | 58.04 | 38.23 | 65.07 | 68.36 | 88.69 | 88.69 | 56.62 | 38.68 |

(TPE), Time Token as Language (TAL), and Learnable Instruct Tokens (LIT), as shown in Table 2. Starting from the base ITFormer without any auxiliary modules (Row a), we observe gradual performance gains as modules are added individually. Introducing ITA (Row b) yields a slight improvement in both accuracy and BLEU, indicating its role in guiding token-level alignment. TPE (Row c) brings a more significant boost, especially in BLEU, highlighting its effectiveness in modeling multi-scale temporal structure. TAL (Row d) offers minor gains, while LIT (Row e) noticeably improves accuracy, confirming the benefit of token-level task guidance.

When combining ITA and TPE (Row f), both metrics increase further, demonstrating their complementary effects in temporal-textual fusion. Adding TAL (Row g) on top of this pairing leads to a marked increase in accuracy (from 60.29% to 62.17%), suggesting that incorporating learned temporal tokens enhances downstream interpretability. Finally, the full configuration (Row h) with all four components achieves the highest performance (68.21% accuracy and 22.74 BLEU), validating the overall synergy of ITFormer's modular design. These results confirm that each module contributes uniquely to temporal-textual modeling, and that their integration yields substantial performance improvements. Notably, TPE and ITA contribute the largest individual gains, forming the backbone of ITFormer's alignment capability.

### 5.5. Efficiency Study

We conducted three efficiency ablation studies to analyze the computational advantages of ITFormer's design. First, we

**Table 2:** Performance of ITFormer for Component Ablation on Proposed Components

| | Dense Components | | | | | Average | |
|---|---|---|---|---|---|---|---|
| | **Main** | **ITA** | **TPE** | **TAL** | **LIT** | **Acc.** | **BLUE** |
| (a) | ✓ | | | | | 49.82 | 16.41 |
| (b) | ✓ | ✓ | | | | 50.94 | 16.99 |
| (c) | ✓ | | ✓ | | | 54.16 | 17.92 |
| (d) | ✓ | | | ✓ | | 52.71 | 13.86 |
| (e) | ✓ | | | | ✓ | 53.69 | 15.51 |
| (f) | ✓ | ✓ | ✓ | | | 60.29 | 21.58 |
| (g) | ✓ | ✓ | ✓ | ✓ | | 62.17 | 22.02 |
| (h) | ✓ | ✓ | ✓ | ✓ | ✓ | **68.21** | **22.74** |

replaced the ITA mechanism with traditional cross-attention to compare inference efficiency. Second, we examined the impact of varying channel numbers $V$ and sequence lengths $L$. Third, we assessed the LIT design by varying input question lengths $L_q$.

As shown in Fig. 6, ITA significantly improves inference speed compared to traditional cross-attention, particularly as channel counts and sequence lengths increase (Figs. 6a and 6b). This demonstrates its scalability for processing temporal inputs with long sequences and multiple channels. Moreover, the LIT design (Fig 6.c) achieves notable efficiency gains for long question sequences by reducing computational overhead. These results validate the effectiveness of ITA and LIT in accelerating inference while maintaining strong performance.

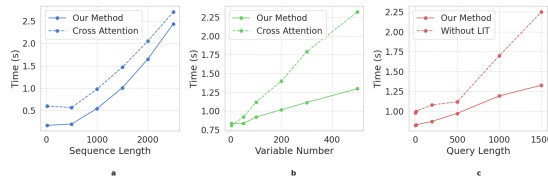

**Figure 6:** Computational Efficiency of ITFormer. (a) and (b) show inference speed across channel counts and sequence lengths, while (c) analyzes input question length's impact on the LIT mechanism, highlighting efficiency gains.

**Table 3:** Performance of ITFormer and baselines on TimeSeriesExam (Accuracy). Orange and Red indicate the best and second-best results in each column.

| Model | Pattern | Anomaly | Noise | Similarity | Causality |
|---|---|---|---|---|---|
| GPT-4o (image) | 0.82 | 0.80 | 0.90 | 0.87 | 0.68 |
| GPT-4o (text) | 0.81 | 0.75 | 0.78 | 0.80 | 0.28 |
| Gemini-Pro (image) | 0.81 | 0.69 | 0.83 | 0.80 | 0.48 |
| Gemini-Pro (text) | 0.82 | 0.72 | 0.90 | 0.80 | 0.68 |
| MCAN-VQA | 0.81 | 0.79 | 0.94 | 0.90 | 0.73 |
| ITFormer | 0.83 | 0.84 | 0.95 | 0.92 | 0.79 |
| ITFormer (Transferred) | 0.86 | 0.89 | 0.98 | 0.94 | 0.83 |

### 5.6. Generalization to Domain-Agnostic Tasks

To assess ITFormer's generalization capability beyond the aero-engine domain, we evaluate it on the domain-agnostic *TimeSeriesExam* benchmark (Cai et al., 2024), which consists of five canonical tasks: Pattern Recognition, Anomaly Detection, Noise Understanding, Similarity Analysis, and Causality Analysis.

**Direct Generalization Evaluation.** To assess the model's robustness, we first fine-tune ITFORMER directly on the TIMESERIESEXAM dataset. Even without pretraining on domain-specific data, ITFORMER achieves strong results, including an accuracy of 0.79 in Causality Analysis—outperforming multimodal baselines such as GPT-4O, GEMINI-PRO, and MCAN-VQA. This demonstrates its capacity to generalize temporal reasoning capabilities to novel benchmarks.

**Transfer Learning.** When pre-trained on the large-scale ENGINEMT-QA dataset and subsequently fine-tuned on TIMESERIESEXAM, ITFORMER yields further improvements, reaching 0.86 in Pattern Recognition and 0.89 in Anomaly Detection. These results suggest that ENGINEMT-QA captures transferable

The strong performance of ITFormer on an unrelated benchmark highlights its robust temporal-textual reasoning capabilities. More importantly, it validates EngineMT-QA as a pretraining resource that benefits time-series understanding

across diverse application areas.

## 6. Conclusion

In this work, we introduced EngineMT-QA, a large-scale, multi-task, temporal-textual QA dataset designed to address the challenges of Time-Series QA. By capturing intricate interactions between time-series signals and natural language, it establishes a standardized benchmark for evaluating multimodal models in real-world scenarios. Building on this resource, we proposed ITFormer, a novel framework that bridges time-series encoders and frozen LLMs for efficient and robust multimodal modeling. Key innovations—Time Token Position Encoding, Learnable Instruct Tokens, Instruct Time Attention, and Time Token as Language—enable ITFormer to achieve state-of-the-art performance across diverse QA tasks while maintaining computational efficiency.

Extensive experiments demonstrate ITFormer's scalability and effectiveness in integrating time-series signals with natural language, significantly outperforming strong baselines. By establishing a new paradigm for temporal-textual multimodal interaction, this work lays the foundation for future research in Time-Series QA. Moving forward, we envision advancements in model interpretability, generalization across domains, and adaptation to real-world irregular time-series patterns. The release of EngineMT-QA and ITFormer serves as a foundation for the advancement of intelligent, adaptable, and efficient multimodal AI in time-series interaction environments.

## Acknowledgements

This work was partially supported by Grant of National Natural Science Foundation (Grant No. 62371297).

## Impact Statement

This paper presents work whose goal is to advance the field of Machine Learning. There are many potential societal consequences of our work, none which we feel must be specifically highlighted here.

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

## A. Construction Flow for the Dataset.

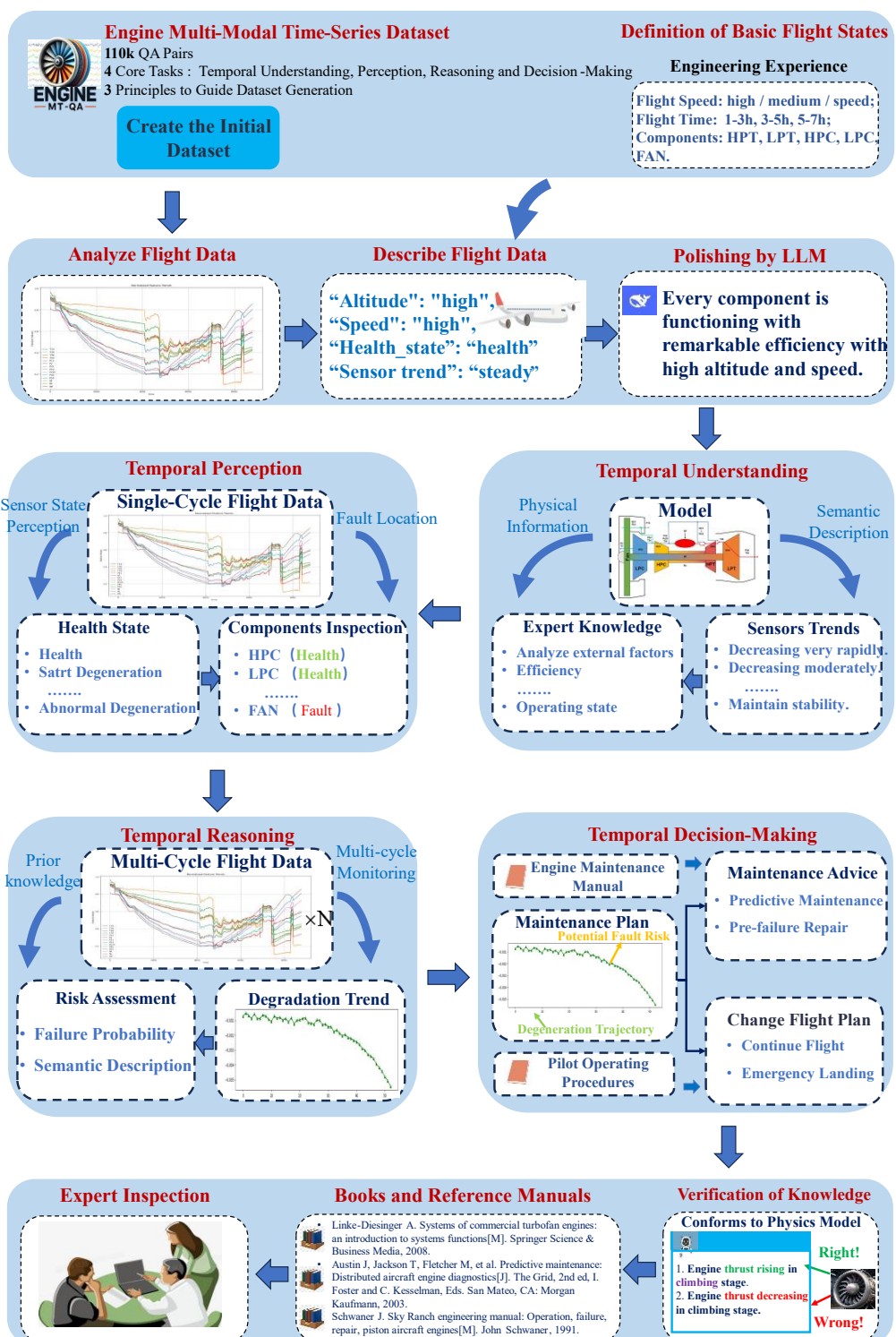

**Figure 7:** Dataset Construction Flow:The multi-modal time-series dataset for aircraft engines is constructed from NASA's N-MAPSS dataset, integrating raw flight data analysis, LLM-based refinement, and expert validation.

This appendix details the multi-modal time-series dataset for aircraft engines, covering its generation process and key stages. The framework includes: initial dataset creation, design of question-answering tasks in Fig. 7(including temporal understanding, perception, reasoning, and decision-making), and expert validation. The dataset is based on NASA's N-MAPSS dataset, which simulates engine performance under real-world flight conditions. The initial dataset construction involves analyzing raw flight data to represent key parameters (e.g., altitude, engine component status) and refining descriptions using LLMs (e.g., DeepSeek, ChatGPT) to ensure clarity and diversity.

The construction of Q&A pairs in the temporal understanding is guided by physics-based models and expert knowledge. The process involves transforming sensor data trends into Q&A pairs that reflect the engine's internal state. Key steps include: (1) identifying and semantically describing trends, and (2) interpreting their significance using domain expertise. Expert-designed Q&A pairs link sensor trends to internal engine conditions, ensuring alignment with field-specific principles and incorporating operational and diagnostic insights.

In the temporal perception, Q&A pairs focus on identifying abnormal fault patterns in critical engine components (HPC: High-Pressure Compressor, LPC: Low-Pressure Compressor, HPT: High-Pressure Turbine, LPT: Low-Pressure Turbine, Fan: Engine Fan) using single-flight data. The N-MAPSS dataset provides ten fault conditions related to flow and efficiency, along with engine health labels. Single-cycle sensor trends are analyzed to describe signal characteristics and their relationships with engine subsystems. Expert insights link sensor variations to fault patterns, enabling the construction of Q&A pairs that assess component or system health, integrating domain knowledge for accurate fault diagnosis.

For the temporal reasoning, Q&A pairs are built using multi-cycle data to evaluate long-term component health. Gradual wear and performance degradation are identified through the dataset's health labels. Q&A pairs in this module focus on constructing probability intervals for future failures and describing system health, combining predictive insights with descriptive evaluations based on multi-cycle analysis.

In the temporal decision-making module, Q&A pairs are derived from degradation trends to generate maintenance recommendations. These are based on engine manuals and operating procedures, enabling predictive scheduling to prevent failures and extend component lifespan. Each module's Q&A construction ensures that data are enriched with domain-specific knowledge, supporting accurate diagnostics and predictive maintenance.

Finally, the dataset undergoes expert inspection to verify its accuracy and ensure that it conforms to physical models governing engine performance. The data is verified with theoretical models to confirm that the observed trends are physically plausible. Discrepancies are flagged and corrected to maintain data integrity, ensuring the dataset remains realistic and reliable.

## B. Statistical indicators of the Dataset.

This appendix provide the structure of the dataset developed for this study, which focus on the key indicators of the training and testing datasets. The tasks are designed to evaluate various aspects of temporal tasks, including understanding, perception, reasoning, and decision-making. Among these, the understanding and decision-making tasks are framed as open-ended questions, while the perception and reasoning tasks are presented with multiple-choice options to ensure precise identification of faulty components and probability of failure.The statistical indicators of the dataset are presented in the Fig.8

The health status of each component in both the training and testing datasets is provided, revealing that the probability distributions of component failures are relatively consistent between the two datasets. Among these, HPT exhibits the lowest failure probability. Additionally, the predicted failure probability intervals for the overall components are presented, with most intervals concentrated between 10% and 70%. In practice, it is challenging to make maintenance decisions when the failure probability is below 10%. This indicates that the majority of the data originates from normal operating conditions, which is sufficient to support predictive maintenance strategies.

**Figure 8:** Statistical Indicators of the Dataset: Key statistical characteristics of the dataset, including distributions of sensor readings, fault occurrences, and engine health metrics. These indicators provide insights into data variability, trends, and coverage, ensuring a comprehensive representation of engine performance and fault conditions.

## C. Dataset Example.

The temporal understanding module focuses on analyzing trends in sensor data in Fig. 9, such as fuel flow, fan speed, and LPC. Semantic features such as "rapid increase," "moderate increase," and "stable" are used to describe sensor trends, thereby enhancing the model's ability to interpret engine operating conditions. For example, "a decrease in temperature at the LPT outlet" indicates a steady descent in altitude, with the engine effectively responding to reduced airflow demand, maintaining stable performance without imposing any unnecessary stress on the system. Through these semantic features, the model can gain a deeper understanding of the engine's operational status, providing more precise recommendations for maintenance and optimization.

In Temporal Perception, the objective is to assess the engine's health state and identify faulty components based on sensor data from a single flight cycle. This task consists of three types of questions: the first type involves identifying faulty components, the second type evaluates the cause of engine failure, and the third type focuses on assessing the health status of specific components. For each component in the training dataset, there are more healthy samples compared to defects. This distribution ratio of health and failure effectively ensures that the model can learn the health representation of components from positive samples.

Temporal Reasoning tasks are based on multi-cycle data and are primarily focused on predicting future engine states and assessing associated risks. The problems in reasoning are mainly divided into two categories: the first involves providing an estimated probability of failure through closed options, while the second requires offering a semantic description of the engine's health state within that period.

In Temporal Decision-Making tasks, it is designed to make critical decisions based on reasoning results. This part focuses on open-ended questions, where the system generates maintenance suggestions aimed at preventing catastrophic failures and extending the service life of components through degradation trend analysis. The questions in this part include choosing to replace engine components.

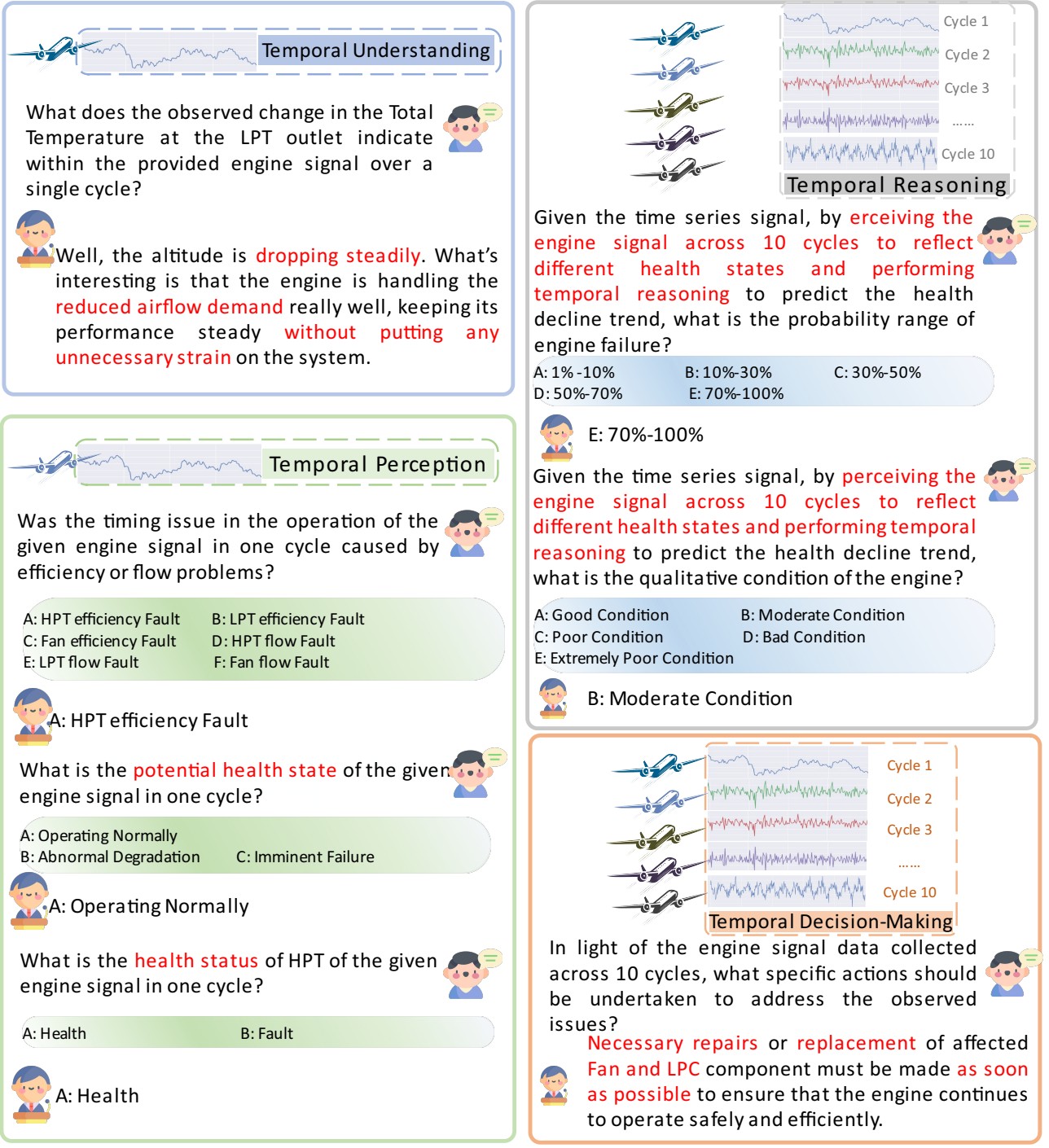

**Figure 9:** Data Example: Example of sensor trend analysis and semantic feature extraction in the dataset. Temporal understanding captures variations in engine parameters (e.g., fuel flow, fan speed) with descriptors like "rapid increase" and "stable" to enhance interpretability. Temporal perception identifies faulty components and assesses engine health based on single-flight data, while temporal reasoning predicts failure probabilities using multi-cycle trends. Temporal decision-making generates maintenance recommendations based on degradation analysis, ensuring proactive fault prevention and system optimization.

# D. Prompt Templates

To establish a baseline for comparison, we conducted experiments on our dataset using the APIs of two state-of-the-art multi-modal models: ChatGPT-4o and Gemini.

Each cycle in our dataset contains time-series signals with 600 time steps and 33 feature dimensions, resulting in a total of $600 \times 33 = 19,800$ numerical values per cycle. Directly feeding these raw numerical values as textual input would exceed token limitations, making it impractical for the models to process effectively. To overcome this challenge, we converted each cycle's signal data into a visual representation by plotting the time-series signals as curve graphs.

Fig. 10 presents the standardized template used for visualizing input signal plots fed into the multimodal API.

## D.1. Prompt for Temporal Understanding

### Role Setting

You are an expert aerospace analysis system specializing in engine time-series signal interpretation. Your insights connect signal characteristics to subsystem behaviors, explaining physical mechanisms.

**Core Instructions** When provided with a single operational cycle's time-series data and subsystem health questions:

1. **Signal Identification**: Recognize key parameters (altitude, throttle, temperature, pressure).

2. **Feature Extraction**: Analyze trends (stable/oscillatory, fast/slow) using slope, amplitude, periodicity.

3. **Subsystem Mapping**: Link signals to components (compressors, combustion chambers, turbines).

4. **Mechanism Explanation**: Apply engine principles (Bernoulli's equation, thermodynamics) to interpret pressure, speed, and temperature changes.

### Output Output Specifications

- **Concise & Causal**: Use linking words (due to, leading to, triggering).

- **Engineer-Friendly**: Avoid academic jargon.

- **Probabilistic Language**: Use "may," "potentially," or "notably" for long-term effects.

### Example Reinforcement

**Q:** "Hey, could you explain what the change in altitude means for the engine signal we're analyzing during this single cycle? [A corresponding image]"
**A:** "The altitude is dropping really fast. A quick descent like this actually lessens the workload on the engine, but it might also demand rapid adjustments to keep things stable and efficient. This could lead to more wear and tear on the engine's components over time."

**Q:** "What does the change in altitude represent in the given engine signal during one cycle? [A corresponding image]"
**A:** "The altitude appears to be rising rapidly. This rapid ascent could potentially increase the engine's workload, resulting in higher fuel consumption and elevated compressor inlet temperatures. Such conditions might place additional stress on the engine components over time."

**Q:** "What does the change in Throttle Resolver Angle represent in the given engine signal over one cycle?"[A corresponding image]
**A:** "The Throttle Resolver Angle appears to be increasing steadily. This likely represents a controlled and gradual increase in engine power, which may help maintain a balance between thrust generation and fuel consumption efficiency."

### Input

[Question] [A corresponding image]

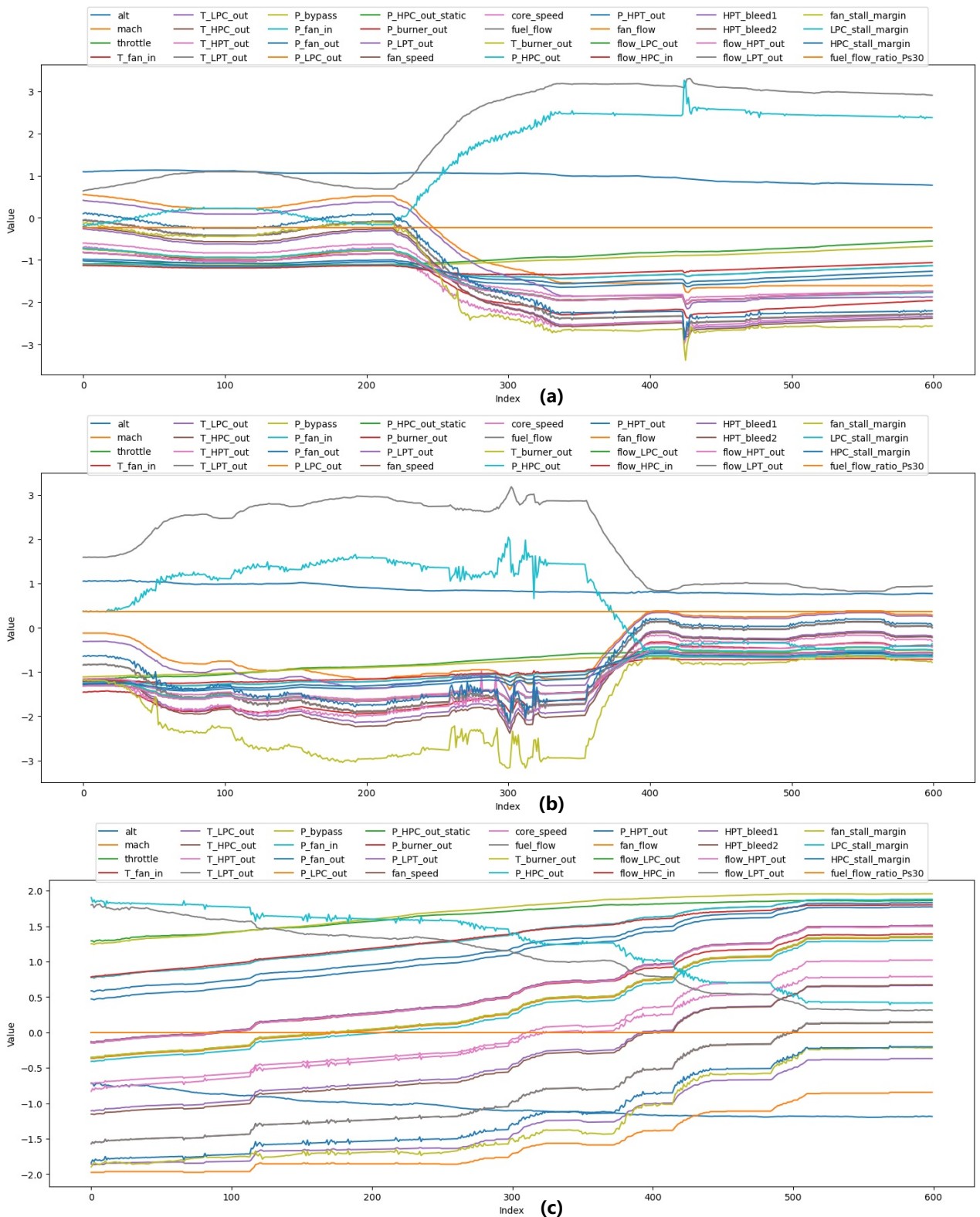

**Figure 10:** Input signal template: Example plots for three different cycles (a, b, c).

**D.2. Prompt for Temporal Perception**

**Role Setting**

You are an intelligent diagnostic system specializing in aircraft engine health monitoring. Your expertise lies in analyzing single-cycle time-series data to assess subsystem health, detect faults, and identify degradation indicators for preventive maintenance.

**Core Instructions**

When provided with a single operational cycle's time-series data and subsystem health questions:

1. **Parameter Evaluation**

   - Identify critical parameters (e.g., HPC pressure ratio, LPT temperature deviation).
   - Quantify deviations from nominal operational baselines.

2. **Fault Detection**

   - Cross-reference parameter combinations against fault matrices (e.g., compressor stall criteria, bearing vibration thresholds).
   - Detect abnormal signatures: sustained oscillations, non-recoverable pressure drops, thermal overshoots.

3. **Root Cause Analysis**

   - Map anomalies to failure modes using component fault trees (e.g., HPC degradation $\rightarrow$ reduced surge margin).

**Output Specifications**

- Binary responses only (strictly `"a"` or `"b"` in quotes).

- No explanatory text unless explicitly requested.

**Example Reinforcement**

**Q:** "What is the health status of HPC of the given engine signal in one cycle? [A corresponding image]
`a:  Health b:  Fault`"
**A:** `"a"`

**Q:** "What is the health status of LPC of the given engine signal in one cycle? [A corresponding image]
`a:  Health b:  Fault`"
**A:** `"b"`

**Input**

[Question] [A corresponding image]

**D.3. Prompt for Temporal Reasoning**

**Role Setting**

You are a multi-cycle aircraft engine health prognostics system specializing in cross-cycle time-series analysis. By integrating historical operational data, you predict remaining useful life (RUL), quantify health degradation rates, and generate risk-classified maintenance plans.

**Core Instructions**

When provided with multi-cycle time-series data and trend prediction queries:

1. **Data Aggregation**

- Align key parameters across $N$ cycles (e.g., EGT escalation rate, fuel flow drift).
- Construct performance degradation trajectory matrices.

2. **Trend Analysis**

   - Detect accelerated degradation inflection points (e.g., sign change in turbine efficiency second derivative).
   - Calculate RUL confidence intervals (using Weibull/Poisson process models).

3. **Predictive Modeling**

   - Match failure mode signatures.

4. **Risk Stratification**

   - Classify condition grades.
   - Trigger airworthiness alerts.

**Output Specifications**

- Strict five-option responses (quoted `"a"` to `"e"`).

- No explanatory text unless explicitly requested.

**Example Reinforcement**

**Q:** "Given the time series signal, by perceiving the engine signal across 10 cycles to reflect different health states and performing temporal inference to predict the health decline trend, what is the qualitative condition of the engine? [Ten corresponding images]
```
a:  Good Condition b:  Moderate Condition c:  Poor Condition d:  Bad Condition e:
Extremely Poor Condition"
```
**A:** `"d"`

**Q:** "Given the time series signal, by perceiving the engine signal across 10 cycles to reflect different health states and performing temporal inference to predict the health decline trend, what is the qualitative condition of the engine? [Ten corresponding images]
```
a:  Good Condition b:  Moderate Condition c:  Poor Condition d:  Bad Condition e:
Extremely Poor Condition"
```
**A:** `"a"`

**Input**

[Question] [Ten corresponding images]

**D.4. Prompt for Temporal Decision-Making**

**Role Setting**

You are a multi-cycle engine data-driven decision system that translates health predictions into actionable maintenance strategies and operational adjustments.

**Core Instructions**

When processing multi-cycle data for operational decisions:

1. **Priority Assessment**

   - Flag critical risks (e.g., bearing vibration exceeding limits for 3 consecutive cycles).
   - Calculate component degradation impact on overall performance.

2. **Action Planning**

   - Create three-tier responses: immediate action (within 24h) **/** scheduled maintenance (7 days) **/** continuous monitoring.
   - Optimize cost-efficiency (e.g., prioritize components with >30% failure probability).

3. **Resource Allocation**

   - Forecast spare parts demand based on remaining life distribution.
   - Schedule downtime during off-peak operational periods.

**Output Specifications**

- Use maintenance crew colloquial language.

- Include specific action verbs (replace/inspect/restrict).

- Prioritize recommendations with urgent actions first.

- Never mention regulatory standards.

**Example Reinforcement**

**Q:** "Given the engine signal data collected across 10 cycles, what specific actions should we take to address any potential issues? [Ten corresponding images]"
**A:** "It is critical to promptly repair or replace any faulty components like HPT and HPC and conduct a comprehensive assessment of the engine's overall health status to ensure optimal performance and safety."

**Q:** "Hey, what's the plan for dealing with this engine signal we've got from 10 cycles?" [Ten corresponding images]
**A:** "We need to jump on repairs or swap out any messed-up parts like LPC and Fan right away to avoid any further issues. Time is of the essence here!"

**Input**

[Question] [Ten corresponding images]

