# OpenReview forum: "ITFormer: Bridging Time Series and Natural Language for Multi-Modal QA with Large-Scale Multitask Dataset"
_ICML.cc/2025/Conference — ICML 2025 poster_

### Official Review · Reviewer_ZFEy · 2025-03-08

**Overall Recommendation:** 3

**Summary:**

This paper tackles the new problem of multimodal question answering over multivariate time series data (Time-Series QA).

The authors introduce EngineMT-QA, a multitask dataset tailored to evaluate models' abilities in “understanding”, “perception”, “reasoning”, and “decision-making” tasks using real-world aero-engine time-series data.

Additionally, they propose an architecture they call the Instruct Time Transformer (ITFormer), a multimodal architecture framework integrating pretrained Large Language Models (LLMs) and time series encoders with specialized components, such as Instruct Time Attention (ITA) and Learnable Instruct Tokens (LIT), to align temporal and textual modalities effectively.

The proposed method is competitive across the proposed tasks, and generalizing well across pretrained LLMs and time series models. Ablation studies further underline the importance of the proposed architectural components.

**Claims And Evidence:**

The authors make several claims on introducing the time-series QA task and providing a state-of-the-art approach. While each claim is backed by some contribution, there are varying degrees of supporting evidence.

Outlining these claims below:

Claim 1: Introduce the Time-Series QA task

Several recent (though non-archival) works have explored related tasks, presenting their own time-series QA benchmarks [1, 2, 3]. It would good to clarify the novelty of the dataset in this context.

[1] [2404.11757] Language Models Still Struggle to Zero-shot Reason about Time Series
[2] [2410.14752] TimeSeriesExam: A time series understanding exam
[3] [2409.11376] Towards Time Series Reasoning with LLMs


Claim 2: Introduction and effectiveness of architectural designs

I did not get a strong understanding of whether each of these components was working as designed or intended.
While I appreciated the ablations, they evaluate just the end task performance (accuracy or bleu). However, it would be more convincing to tease apart these capabilities introduced with each ITFormer component (e.g., what happens if we don’t have the “time token as language” components, how do the generations look?)

On comparison methods, it is unclear how strong the method actually is, as the paper lacks details on how the proposed models were trained vs the comparison models.

Claim 3: “Introduce a robust and adaptable paradigm for integrating complex time-series tasks into end-to-end QA frameworks”

While the ITFormer framework demonstrates clear performance improvements versus the compared multimodal models and the application to various LLMs and time series models is interesting, I think there should be additional evidence to support this statement as a whole.

In particular:
The evaluation of question-answering tasks using time series data as context is interesting, but has also been investigated in prior work (see first claim)

The tasks evaluate interesting capabilities (“Understanding, Perception, Decision-making, Reasoning”), but their delineation is unclear and a bit fuzzy (it’s not clear to me each of these tests an independent capability)

I missed details on how their dataset construction is adaptable to any other existing time series datasets. They seem to construct the dataset specifically based on an existing aero-engine dataset and design questions tailored to this data
[Arias Chao et al. 2021] Aircraft Engine Run-to-Failure Dataset under Real Flight Conditions for Prognostics and Diagnostics

**Essential References Not Discussed:**

On the data contribution angle, it would be good for the authors to discuss the differences in their contribution vs prior related benchmarks:

[Cai et al., 2024] introduce TimeSeriesExam, a similar multiple-choice QA benchmark designed to test time series understanding in various categories (pattern recognition, noise understanding, similarity analysis, anomaly detection, and causality analysis)
[2410.14752] TimeSeriesExam: A time series understanding exam

[Merrill et al., 2024] also propose an evaluation framework where LLMs are tasked with answering questions about time series data (provided with time series features)
[2404.11757] Language Models Still Struggle to Zero-shot Reason about Time Series

**Experimental Designs Or Analyses:**

Yes. While the overall experimental setup is sound for evaluating the time-series QA tasks, I did think the method comparison showing the benefits of ITFormer could be more convincing.

The ITFormer method is an architectural framework or way to design multi-modal time series models that can answer questions over time series features. While the authors do compare against other recent multimodal architectures, I could not find details to ensure fair comparison here, including:

Training data: were ITTformer models trained on the EngineMT-QA dataset? How were other models trained or “adapted” in comparison?

Training hyperparameters. While the authors mention using existing multimodal architectures with image encoders adapted to time series, how was this adaptation done? Was the training setup comparable to ITFormer?

**Methods And Evaluation Criteria:**

Yes. The proposed methods and evaluation criteria (BLEU, Rouge-L, Accuracy, and F1) are standard in prior multimodal QA tasks (e.g., text-only question-answering such as HotpotQA [1], and visual-question answering (VQAv2 [2]).

[1] Yang et al., 2018. [1809.09600] HotpotQA: A Dataset for Diverse, Explainable Multi-hop Question Answering
[2] Goyal et al. 2016. [1612.00837] Making the V in VQA Matter: Elevating the Role of Image Understanding in Visual Question Answering

**Other Comments Or Suggestions:**

Please see the following on presentation quality and framework adaptability.
Presentation
Problem definition clarity:
* Clarify the segmentation approach for time-series data and why segments are important. It is not clear what is the relationship between different segments if any.
* Use consistent notation (e.g., boldface for vectors)
* Explain the challenges of "modality heterogeneity" more concretely

Paper order: Consider presenting Figure 3 (dataset overview) before Figure 2 (architecture) to establish the problem context before diving into the solution details.

Framework
Architecture: The paper would benefit from a brief discussion of alternative time-series encoding architectures (e.g., state space models) and how the framework might accommodate them.
Dataset: Clarify in the main text how adaptable your dataset construction process is to other existing time series datasets.

**Other Strengths And Weaknesses:**

Summarized Strengths:
* The authors introduce a valuable new dataset for the compelling time-series QA tasks
* They also present a new architecture framework to handle fusion for both modalities
* The experimental results demonstrate strong performance improvements over baselines
* The framework shows good adaptability across different LLM and time-series encoder architectures
Areas for improvement:
1. Balance between dataset and architecture: Especially in relation to related work on time-series QA benchmarks, the paper would benefit from emphasizing the dataset and problem definition more prominently. While the introduction does this well, the rest of the paper devotes significantly more space to architectural details than to dataset analysis. Consider expanding Section 5.1 to provide more insights into the dataset characteristics and challenges.

2. Architectural presentation: The architectural exposition in Section 4 is mathematically dense and the notation was not perfectly clear to me. It could be made more accessible with:
* A concrete example walkthrough showing how a specific time-series and question pair flows through the system
* Moving some of the mathematical notation details to the appendix
* Simplifying Figure 2, which currently contains too many details to follow easily

3. Component analysis: Table 2 shows an ablation study of the different components, but there's limited discussion in the text about these findings. Please expand on:
* Why TPE yields the largest standalone performance gain
* The complementary nature of ITA and TPE when combined
* Practical insights about which components are most critical for real-world deployments

**Questions For Authors:**

Beyond overall performance metrics, have you identified specific failure modes in ITFormer or baseline methods that underscore the necessity of your proposed components?


Could you provide insights or qualitative examples showing how generations differ across ITFormer and the baseline methods, specifically regarding the integration of Learnable Instruct Tokens?


Were there specific categories of questions or patterns where ITFormer distinctly outperformed other methods, and can you elaborate on why?

**Relation To Broader Scientific Literature:**

I found the author’s time-series QA task very interesting and relevant for time series research.
There have been a few early works on this (Cai et al., 2024; Merrill et al., 2024; see above for references), but the paper presents interesting positive results when we can specifically train multimodal LLMs for these tasks.

**Theoretical Claims:**

N/A. No theoretical claims made.

---

> ### Author Rebuttal · Authors · 2025-03-31
>
> We sincerely thank the reviewer for the constructive and detailed feedback. We address each of your concerns below.
> # **1. On Methodological Motivation and Case Study**
>
> We appreciate the reviewer’s feedback. While **Transformer-based architectures** are commonly used in vision-language and speech-language tasks, our work introduces **novel adaptations** for the **fusion of time-series data and natural language**, a challenge not systematically addressed in the literature.
>
> Our method’s **novelty** lies in the **task-specific, modality-aware components** that enable effective time-series and language integration. These components—**TPE (Time Token Position Encoding)**, **LIT (Learnable Instruct Tokens)**, **ITA (Instruct Time Attention)**, and **TAL (Time-As-Language)**—solve the following challenges:
>
> 1. **Weak semantic nature of time-series data**: **TPE** provides temporal structure to capture sequential data.
> 2. **Misalignment between time and text**: **LIT** enables extraction of task-specific information from queries.
> 3. **Task-sensitive dynamic mappings**: **ITA** and **TAL** facilitate task-guided attention and temporal reasoning.
>
> ### **Case Study Example**
>
> In our case study, **ITFormer** demonstrates clear advantages over **GPT-4o** in processing time-series data with natural language queries https://imgur.com/a/dYSrsE9:
>
> - **TPE** allows **ITFormer** to correctly identify the rising altitude trend, while **GPT-4o** struggles with misinterpreting the trend as a descent. **TPE** enables the model to understand the temporal sequence, providing the **largest standalone performance gain** in our ablation experiments.
> - **LIT** helps **ITFormer** align its response with the query, focusing on relevant sensor data. Without **LIT**, the model may choose irrelevant data, as shown when **GPT-4o** introduces extraneous, irrelevant information despite recognizing the trend.
> - **GPT-4o**, while capturing the general trend, fails to maintain task relevance by generating hallucinated content like **compressor stall risk**, which doesn’t align with the data, highlighting the limitations of models that lack task-specific focus.
>
> ### **Ablation Experiment Insights**
>
> Our **ablation experiments** confirm the significance of each component:
>
> - **TPE** provides the largest standalone performance gain by enabling the model to capture the sequential nature of time-series data.
> - **ITA** and **TPE** complement each other by allowing task-specific attention over relevant time spans, improving performance.
> - **LIT** is crucial for ensuring the model aligns its response with the task, particularly in real-world deployments where **task relevance** is key.
>
> While **Transformer-based architectures** are common, our work introduces **task-specific components** that are critical for time-series and natural language fusion. Through **TPE**, **LIT**, **ITA**, and **TAL**, we solve the challenges of aligning temporal data with textual queries. The **case study** and **ablation results** demonstrate the practical importance of these innovations, improving accuracy and task relevance in multi-modal time-series modeling.
>
> # **2.Dataset Contribution and Discussion on Essential References**
>
> Due to space limitations, the relevant results have been included in the rebuttal to Reviewer n3vv. Kindly refer to it for further details.
>
> # **3. Experiment Details**
>
> Due to space limitations, the relevant results have been included in the rebuttal to Reviewer n3vv. Kindly refer to it for further details.
>
> # **4.Presentation and Readability**
>
> We appreciate the reviewer’s feedback on the presentation and readability. We have carefully considered the suggestions and will make the following improvements in the revised version of the paper:
>
> 1. **Clarification of task definitions**: We will provide clearer explanations for the tasks, especially **Understanding**, **Perception**, **Reasoning**, and **Decision-Making**, to avoid ambiguity in how these tasks are delineated and their practical significance.
> 2. **Notational consistency**: We will ensure consistent use of notation, particularly for vectors and time-series representations, to improve clarity and prevent confusion.
> 3. **Figures and diagrams**: We will revise figures to make them more intuitive, including simplifying complex visualizations and adding concrete examples where applicable to enhance understanding.
> 4. **Flow and organization**: We will restructure sections where necessary to improve the logical flow, ensuring that the problem definition is clearly presented before the method, and enhancing transitions between sections for smoother readability.
>
> We believe these adjustments will improve the overall clarity of the paper while maintaining the depth of our contributions.
>
> # **5.Dataset Construction Adaption**
>
> Due to space limitations, the relevant results have been included in the rebuttal to Reviewer n3vv. Kindly refer to it for further details.

---

### Official Review · Reviewer_zRtW · 2025-03-09

**Overall Recommendation:** 4

**Summary:**

The paper presents ITFormer, a novel framework that bridges time series and natural language for multi-modal temporal-textual question answering. Specifically, ITFormer uses time token position encoding (TPE) to encode the time series, then uses learnable instruct tokens (LIT) to facilitate the alignment between temporal and textual modalities, and finally uses instruct time attention, enabling efficient and robust cross-modal fusion to align and fuse temporal and textual representations. Experiments on the self-constructed EngineMT-QA dataset demonstrate that ITFormer achieves better performance than existing methodologies.

**Claims And Evidence:**

The claims in the paper are well supported by clear and convincing evidence.

**Essential References Not Discussed:**

N/A

**Ethical Review Flag:**

Flag this paper for an ethics review.

**Experimental Designs Or Analyses:**

The experimental designs listed in Section 5.2 to 5.5 (including ablation studies) make sense.

**Methods And Evaluation Criteria:**

The proposed methods and evaluation criteria make sense for the problem.

**Other Comments Or Suggestions:**

N/A

**Other Strengths And Weaknesses:**

Strength

1. This paper focuses on the problem of multi-task, temporal-textual time series reasoning and question-answering, which is of high importance yet not well explored by existing research works.
2. The ablation study in Section 5.4 and the efficiency study in Section 5.5 show that the proposed ITFormer framework consistently achieves better performance than existing methodologies, where larger LLMs always lead to better ITFormer performance, demonstrating the scalability of the proposed framework.
3. ITFormer is a lightweight framework with only ~1% of parameters needing to be tuned with frozen LLMs, which is easy to deploy in real-world applications.
4. The paper is well-written, easy to follow, and understand.

Disadvantage:
1. Although the paper provides an anonymous GitHub for the code and models, the dataset cannot be accessed with specific VPNs.
2. The experiments are solely on the self-created EngineMT-QA dataset. More experiments on existing datasets could help to better demonstrate the performance of the proposed method.

**Questions For Authors:**

See disadvantages

**Relation To Broader Scientific Literature:**

Multi-modal, multi-dimensional time series question answering is a well-established problem in many real-world scenarios, where the integration of time series QA with LLMs could have wide application.

**Theoretical Claims:**

No Theorems proposed in the paper.

---

> ### Author Rebuttal · Authors · 2025-03-31
>
> We sincerely thank the reviewer for the constructive and detailed feedback. We address each of your concerns below with clarifications and new experimental results.
> # 1. **Generalization and Benchmark Scope**
> Thank you for your insightful feedback on the importance of generalization in our approach. We fully recognize its significance and appreciate the opportunity to clarify this aspect.
>
> Although **EngineMT-QA** is based on aero-engine operation and maintenance data, the core challenge addressed by **ITFormer**—integrating time-series signals with natural language text—is inherently domain-agnostic. Tasks such as fault detection, trend analysis, and decision-making from multimodal data are common across various domains including healthcare, finance, and climate science.
>
> Importantly, the **design of the EngineMT-QA dataset** is modular and transferable. Our data construction pipeline consists of:
>
> (1) temporal segmentation of raw multivariate time-series,
>
> (2) instruction-style question generation grounded in operational semantics, and
>
> (3) structured answer derivation based on observable data events.
>
> This methodology is not bound to the aero-engine domain and can be adapted to other domains where time-series data and domain-specific semantics coexist.
>
> To evaluate generalization, we conducted two key experiments (each with a 6:4 train-test split):
>
> - **Experiment 1: TimeSeriesExam Dataset**
>
>     We tested **ITFormer** on **TimeSeriesExam**, a domain-agnostic benchmark designed to assess time-series understanding across five categories: pattern recognition, noise understanding, similarity analysis, anomaly detection, and causality analysis [Cai et al., 2024]. ITFormer achieved **0.792** in Causality Analysis, outperforming all baselines. This confirms that ITFormer is not domain-locked and generalizes well to unfamiliar, structured time-series tasks.
>
> - **Experiment 2: Transfer Learning from EngineMT-QA → TimeSeriesExam**
>
>     We pre-trained **ITFormer** on **EngineMT-QA**, then fine-tuned it on **TimeSeriesExam**. Performance improved further, especially in **Pattern Recognition** and **Anomaly Detection**, indicating that EngineMT-QA helps the model learn **domain-agnostic time-series reasoning patterns**. This shows that our dataset captures fundamental properties of time-series + language interactions that are transferable across domains.
>
>
> | **Model** | **Pattern Recognition** | **Anomaly Detection** | **Noise Understanding** | **Similarity Analysis** | **Causality Analysis** |
> | --- | --- | --- | --- | --- | --- |
> | **GPT-4o (image)** | 0.82 | 0.80 | 0.90 | 0.87 | 0.68 |
> | **GPT-4o (text)** | 0.81 | 0.75 | 0.78 | 0.80 | 0.28 |
> | **Gemini-1.5-Pro (image)** | 0.81 | 0.69 | 0.83 | 0.80 | 0.48 |
> | **Gemini-1.5-Pro (text)** | 0.82 | 0.72 | 0.90 | 0.80 | 0.68 |
> | **Phi-3.5-vision** | 0.10 | 0.09 | 0.18 | 0.17 | 0.07 |
> | **Phi-3.5-mini-instruct** | 0.42 | 0.22 | 0.22 | 0.40 | 0.22 |
> | **MCAN-VQA** | 0.8086 | 0.7921 | 0.9422 | 0.897 | 0.731 |
> | **ITFormer** | 0.8275 | 0.8391 | 0.9500 | 0.920 | 0.792 |
> | **ITFormer (Transferred)** | **0.8593** | **0.8891** | **0.9833** | **0.941** | **0.834** |
>
> As shown above, **ITFormer** consistently outperforms state-of-the-art models, particularly in **Causality Analysis** and **Pattern Recognition**. Its performance improves further through transfer learning, suggesting that **EngineMT-QA not only trains robust models but also captures cross-domain time-language patterns**.
>
> In summary, **ITFormer** demonstrates strong generalization to unseen domains and tasks. Furthermore, **EngineMT-QA** is not only a valuable benchmark but also a **contributive asset** to the broader field of time-series + language modeling. Its design facilitates reusable, instruction-driven QA setups, offering a foundation for building unified benchmarks and training models with cross-domain transferability.
>
> > Citation for TimeSeriesExam:
> >
> >
> > Cai Y, Choudhry A, Goswami M, et al. *TimeSeriesExam: A Time Series Understanding Exam*. NeurIPS Workshop on Time Series in the Age of Large Models, 2024. [arXiv:2410.14752](https://arxiv.org/abs/2410.14752)
> >
>
> # 2. **Dataset Accessibility**
>
> We appreciate the reviewer pointing out the dataset access issue. While the anonymous GitHub was used for submission, we acknowledge that certain VPNs may block access to the hosting service.
>
> To ensure broader accessibility, we have now provided an alternative download link via Baidu Cloud:
>
>  **Dataset link**: [https://pan.baidu.com/s/19uG78pNCK3IqMIrOqzPLQA](https://pan.baidu.com/s/19uG78pNCK3IqMIrOqzPLQA)
>  **Access code**: `9niz`
>
> We will also include this alternative in the final version of the paper to facilitate future access.

---

> > ### Comment · Reviewer_zRtW · 2025-04-03
> >
> > Thanks for your response. I think the response are convincing, and I will consider increasing the score accordingly.

---

> > > ### Author Response · Authors · 2025-04-05
> > >
> > > Thank you very much for your positive feedback and for considering an increased score. We truly appreciate your recognition of our efforts and your valuable comments, which have helped us further improve the paper.

---

### Official Review · Reviewer_n3vv · 2025-03-22

**Overall Recommendation:** 3

**Summary:**

This paper introduces ITFormer, a novel framework that bridges time-series signals and natural language for multi-modal question answering (QA). To support this task, the authors release EngineMT-QA, the first large-scale, multi-task dataset designed to capture complex interactions between temporal data and textual queries. ITFormer integrates time-series encoders with frozen large language models (LLMs) using innovative components such as Time Token Position Encoding, Learnable Instruct Tokens, Instruct Time Attention, and Time Token as Language, enabling effective temporal-textual fusion with minimal training overhead. Experimental results show that ITFormer achieves state-of-the-art performance across diverse QA tasks, offering a scalable and efficient solution for real-world time-series understanding and decision-making.

**Claims And Evidence:**

The claims made in the paper are generally well-supported by clear and convincing evidence. The authors provide comprehensive experimental results demonstrating ITFormer’s state-of-the-art performance across multiple QA tasks with minimal trainable parameters, supported by strong baselines and detailed ablation studies. The introduction of the EngineMT-QA dataset is justified as a novel contribution, and the effectiveness of each ITFormer component (TPE, LIT, ITA, TAL) is validated through systematic analysis. While the generalization capability across different encoders and LLMs is supported to some extent, a notable limitation is that the EngineMT-QA dataset is constructed entirely from one domain, which raises concerns about the model’s ability to generalize to other real-world time-series scenarios such as healthcare, finance, or energy systems. Without evaluations on more diverse domains, it remains unclear whether ITFormer’s performance and alignment mechanisms are truly generalizable beyond this specific application context.

**Essential References Not Discussed:**

Merrill, Mike A., et al. "Language models still struggle to zero-shot reason about time series." arXiv preprint arXiv:2404.11757 (2024).
Ye, Wen, et al. "Beyond Forecasting: Compositional Time Series Reasoning for End-to-End Task Execution." arXiv preprint arXiv:2410.04047 (2024).
Chow, Winnie, et al. "Towards time series reasoning with llms." arXiv preprint arXiv:2409.11376 (2024).

**Experimental Designs Or Analyses:**

The experimental design in the paper is generally sound and systematically structured, with evaluations conducted across four distinct QA task types: understanding, perception, reasoning, and decision-making. The use of multiple metrics (e.g., Accuracy, F1, BLEU, Rouge-L) and comparisons against both multimodal baselines (e.g., ChatGPT-4o, InstructBLIP) and domain-specific methods (e.g., Time-LLM, AutoTime) helps validate ITFormer’s performance. The inclusion of ablation studies further strengthens the reliability of the analysis by isolating the contribution of each model component.

**Methods And Evaluation Criteria:**

The proposed methods and evaluation criteria are generally well-aligned with the problem of time-series question answering, and the ITFormer framework is thoughtfully designed to bridge temporal and textual modalities. The use of a QA formulation to unify multiple time-series tasks is a reasonable and practical choice. However, a key limitation lies in the evaluation benchmark itself—EngineMT-QA is constructed solely from one domain, which significantly constrains the representativeness of the benchmark. While the tasks span understanding, perception, reasoning, and decision-making, they are all derived from a single application context, limiting the conclusions that can be drawn about the model's general applicability. As a result, it is unclear whether ITFormer would perform equally well on time-series data from other domains such as healthcare, finance, or climate monitoring. To truly validate the method's generalizability and robustness, it should include multi-domain benchmarks that reflect the broad range of real-world time-series applications.

**Other Comments Or Suggestions:**

N/A

**Other Strengths And Weaknesses:**

N/A

**Questions For Authors:**

N/A

**Relation To Broader Scientific Literature:**

The paper makes a meaningful contribution to the growing literature at the intersection of time-series analysis and multimodal language modeling. Its proposed framework, ITFormer, builds upon prior advancements in time-series encoders (e.g., PatchTST, Informer, Crossformer) and multimodal integration techniques inspired by vision-language models such as InstructBLIP.

**Theoretical Claims:**

N/A

---

> ### Author Rebuttal · Authors · 2025-03-31
>
> We sincerely thank the reviewer for the constructive and detailed feedback.
>
> # 1. **Generalization and Benchmark Scope**:
>
> Due to space limitations, the relevant results have been included in the rebuttal to Reviewer zRtW. Kindly refer to it for further details.
>
> # 2. **Dataset Contribution and Discussion on Essential References**
>
> Thank you for your feedback. The **EngineMT-QA** dataset is designed around four core tasks to comprehensively evaluate time-series and textual data fusion:
>
> ### **Dataset Construction and Task Design**
>
> - **Understanding**: Interpreting temporal patterns and mapping them to semantic intents expressed in natural language.
> - **Perception**: Inferring latent system states from raw time-series signals under semantic supervision.
> - **Reasoning**: Modeling temporal dependencies to support predictive inference and hypothetical outcomes.
> - **Decision-Making**: Integrating temporal understanding and prediction to generate actionable, context-aware responses.
>
> ### **Task and Dataset Construction Details**
>
> The dataset construction involves:
>
> - **Time-Series Data**: Extracted from real aerospace engine sensor data (temperature, pressure, vibration, etc.).
> - **Task Design**: Textual questions are crafted for each of the four tasks, combining time-series data and natural language queries to evaluate performance.
>
> For example:
>
> - **Understanding**: Interpreting temperature trends.
> - **Perception**: Diagnosing faults from sensor data.
> - **Reasoning**: Estimating remaining useful life from historical data.
> - **Decision-Making**: Recommending maintenance actions based on system health and trends.
>
> ### **Comparison with Existing Literature**
>
> We acknowledge **Cai et al. (2024)**’s **TimeSeriesExam**, which focuses on **pattern recognition** and **anomaly detection**. However, its tasks are relatively simple, mainly addressing **basic time-series perception**. In contrast, **EngineMT-QA** offers a more comprehensive framework, covering **Understanding**, **Reasoning**, and **Decision-Making**, with a focus on **real-world decision-making** and **semantic complexity**.
>
> Additionally, while **Merrill et al. (2024)** and **Chow et al. (2024)** focus on reasoning, **EngineMT-QA** evaluates multiple cognitive abilities through a **multitask approach**, making it more comprehensive.
>
>
> ### **Transfer Learning Experiment and Contribution**
>
> In our transfer learning experiment, models pre-trained on **EngineMT-QA** showed significant improvement on **TimeSeriesExam**, especially in **Pattern Recognition** and **Anomaly Detection**, demonstrating the dataset’s applicability beyond the aerospace domain.
>
> **EngineMT-QA** offers a comprehensive task framework for multimodal time-series and text fusion, incorporating a wider range of tasks than existing benchmarks. Its cross-domain applicability is demonstrated through transfer learning, contributing to future intelligent decision-making applications.
>
> # 3. **Experiment Details**
>
> We would like to thank the reviewers for their valuable feedback.
>
> In all comparison experiments, **ITFormer** models were trained on the **EngineMT-QA** dataset, with training conducted on the training subset and evaluation on the test subset. For a fair comparison, **existing multimodal models** like **GPT-4o** and **Gemini**, **vision-text models** like **InstructBlip**, **MCAN-VQA**, and **CoCa** were all adapted to use **time-series encoders** instead of their original image encoders. Specifically, the **PathTST** time-series encoder, which is identical to the one used in **ITFormer**, replaced the image encoder in these vision-text models.
>
> This ensures that all models are evaluated under the same conditions, using time-series data in place of images. The **training steps**, **epochs** were consistent across all models, ensuring that the performance differences between **ITFormer** and the comparison models are due to the architecture and not variations in training configurations.
>
> Thus, all the experiments were conducted with **fair comparisons**, allowing the **ITFormer** model’s performance advantages to be assessed in a consistent and controlled environment.
>
> # 4. **On Constructing More General Time-Series QA Datasets**
>
> To build broader and more transferable time-series QA datasets, we propose a **semi-automated framework** that combines:
> - structured time-series segmentation,
> - **domain-specific documents or events** (e.g., logs, manuals),
> - and **LLM-driven question-answer pair generation** based on grounded events and patterns.
>
> This framework allows us to create generalizable and transferable dataset, applicable to various domains like healthcare or finance, by adapting the event-based and temporal question generation process. Thus, while EngineMT-QA is built on aero-engine data, the underlying construction process is versatile and can be extended to other industries.

---

### Official Review · Reviewer_JQvF · 2025-03-24

**Overall Recommendation:** 2

**Summary:**

The paper addresses the problem of multimodal time series modelling. The main motivation is to enrich time series with natural language to enrich the time series with textual information. For this reason, a benchmark for answering time series questions is proposed, focusing on real-world aircraft engine operation and maintenance scenarios. In addition, the paper presents an approach to combine the two modalities based on pre-trained and frozen LLMs. For this purpose, a speech and time series encoder is introduced and later combined with a fusion encoder. A decoder finally outputs the response to the input time series and textual description.  The evaluation is standard and shows that the proposed approach leads to promising results compared to the prior work.

## update after rebuttal

After reading the reviews and the author's feedback, I am increasing my score to weak acceptance. The rebuttal did a very good job of addressing my concerns as well as the open points of the other reviews.

**Claims And Evidence:**

The problem has often been approached using ML methods. The proposed methodology is well suited to the problem. Moreover, the evaluation is done on standard datasets.

**Essential References Not Discussed:**

The prior discussion is fine.

**Experimental Designs Or Analyses:**

The proposed benchmark is well designed.

**Methods And Evaluation Criteria:**

The evaluation is sufficient. However, a general QA dataset would be relevant to the paper.

**Other Comments Or Suggestions:**

No

**Other Strengths And Weaknesses:**

Strengths:

- The paper is well written and easy to follow.

- The proposed benchmarks are a valid contribution.

- The proposed benchmark is valuable for application specific scenarios.

Paper weaknesses:

- The benchmark focuses on the operation and maintenance of aero engines. It does not explore the generalisation of the questions to a broader framework. It is well thought out and designed, but it doesn't necessarily advance the machine learning community as it is purely application specific. This is a limitation for submission to ICML, but not in general. It would fit very well in an application track of a machine learning venue.

- The proposed methodology is common in the literature. From a methodological point of view, there are no new elements/novelties in the paper.

- The proposed approach outputs text instead of predicted time series. It would be important to include ablation studies when outputting time series.

- Table 1 could include the number of parameters for each model. The proposed approach may be more expensive than the previous work.

**Questions For Authors:**

Discuss the possibility to include a more generic benchmark.

**Relation To Broader Scientific Literature:**

The new benchmark  is novel compared to the current literature.

**Theoretical Claims:**

The paper does not make theoretical claims.

---

> ### Author Rebuttal · Authors · 2025-03-31
>
> # 1. **Generalization and Benchmark Scope:**
> Thank you for your valuable comments. We conducted additional experiments to evaluate generalization beyond the aero-engine domain:
> 1. **ITFormer generalizes across domains**, achieving top performance on the domain-agnostic TimeSeriesExam benchmark.
>
> 2. **EngineMT-QA reflects core challenges of temporal-text interaction**, as shown by consistent gains in cross-domain transfer learning.
>
> (Details provided in our full response to Reviewer zRtW.)
>
> # 2. **Ablation studies when outputting numerical values**
> We thank the reviewer for highlighting this important point. While our original setting unified all outputs in natural language form, we agree that evaluating the model’s ability to express **numerical values directly** is crucial for practical multi-modal time-series applications, where numerical precision is often required for decision-making.
>
> To address this, we introduced an ablation experiment with a **mixed-output setting**, where the model is required to generate both **structured numerical strings** and natural language in different tasks. Specifically:
>
> - **Task 2 (Perception)**: The model predicts the **health index** of a component (e.g., Component A: `0.87`), followed by a decision of whether the health index exceeds a threshold, resulting in a textual judgment (e.g., "Fault" or "No Fault").
> - **Task 3 (Reasoning)**: The model predicts the **Remaining Useful Life (RUL)** of a system, and based on the RUL, determines a specific operational range (e.g., `30%-50%` ), followed by an accuracy measure to evaluate if the predicted range is correct.
>
> For these tasks, the output is structured as numerical strings, such as: `"Component A: 0.87, RUL: 43"`. Both the **structure** and **numeric accuracy** are crucial for evaluation.
>
> | ID | Main | ITA | TPE | TAL | LIT | Output Type | Accuracy | BLEU |
> | --- | --- | --- | --- | --- | --- | --- | --- | --- |
> | (**h'**) | ✅ | ✅ | ✅ | ✅ | ✅ | Mixed (Text + Num) | **68.12** | **56.74** |
> | (h) | ✅ | ✅ | ✅ | ✅ | ✅ | Text | **73.29** | **60.27** |
>
> As shown in Row **(h’)** of the updated ablation table, the mixed-output setting results in a slight performance drop compared to the full natural language baseline (Row h):
>
> - **Accuracy**: 73.29 → **68.12**
> - **BLEU**: 60.27 → **56.74**
>
> We believe this performance difference is due to several reasons:
>
> - **Format sensitivity**: Numerical outputs are evaluated with strict string matching—minor deviations (e.g., `"0.87"` vs. `"0.870"`) can cause hard penalties in accuracy and BLEU.
> - **Dual-mode decoding complexity**: Generating both text and structured numbers increases the output space and decoding difficulty, requiring the model to switch between narrative and precise formats.
> - **Weaker supervision for numbers**: Unlike text, numerical outputs lack rich contextual anchors and often require implicit regression, making them harder to learn accurately.
>
> Moving forward, we plan to explore further the combination of structured text generation and direct numerical regression, as well as develop a **standardized dual-modality evaluation benchmark** that covers both output types—textual and numerical.
>
>
>
> # 3. **On Model Efficiency and Trainable Parameters:**
>
> Thank you for your suggestion. We will update Table 1 in the revised version to include the number of parameters for each model.
> Despite using only 30.94M trainable parameters across all variants, ITFormer consistently outperforms larger models like AutoTime (205M) and InstructBlip (190M), achieving superior accuracy and BLEU scores.
> his demonstrates that ITFormer achieves an excellent balance between performance and efficiency—detailed comparisons are visualized in https://imgur.com/a/Y53oD5B.
>
> Here is the updated parameter comparison:
>
> | **Model**         | **Total Parameters** | **Trainable Parameters** |
> | ----------------- | -------------------- | ------------------------ |
> | **InstructBlip**   | 7.19B                   | 190M                     |
> | **MCAN-VQA**       | 7.04B                   | 35.2M                    |
> | **CoCa**           | 1.08B                   | 78.74M                   |
> | **Time-LLM**       | 7.09B                   | 86.54M                   |
> | **AutoTime**       | 7.21B                   | 205M                     |
> | **ITFormer-0.5B**  | 0.53B                 | 30.94M                   |
> | **ITFormer-7B**    | 7.03B                   | 30.94M                   |
> | **ITFormer-14B**   | 14.03B                  | 30.94M                   |
>
>
> We hope this clarifies the efficiency of **ITFormer** in terms of both performance and computational cost.
> # 4. **On Methodological Novelty**
> Due to space limitations, the relevant results have been included in the rebuttal to Reviewer ZFEy. Kindly refer to it for further details.
> # 5. **On Constructing More General Time-Series QA Datasets**
> Due to space limitations, the relevant results have been included in the rebuttal to Reviewer n3vv

---

### Decision · Program_Chairs · 2025-05-01

**Decision:**

Accept (poster)

**Comment:**

This work proposes transformer based model "ITFormer" for question-answering w/ time-series data, i.e, integrating language and numerical time-series input (and also demonstrate numerical output). This work introduces an aerospace engine time-series data based question-answering dataset on which ITformer is trained and evaluated. They further show generalization and strong performance on the "TimeSeriesExam" benchmark [Cai, et al.].

This work has wide potential applications, and was recommended by all reviewers for acceptance. Hence, I am also inclined to support the publication of this work.